# FIRA: CAN WE ACHIEVE FULL-RANK TRAINING OF LLMS UNDER LOW-RANK CONSTRAINT?

## ABSTRACT

Low-rank training has emerged as a promising approach for reducing memory usage in training Large Language Models (LLMs). Previous methods either rely on decomposing weight matrices (e.g., LoRA), or seek to decompose gradient matrices (e.g., GaLore) to ensure reduced memory consumption. However, both of them constrain the training in a low-rank subspace, thus inevitably leading to sub-optimal performance. This raises a question: whether it is possible to consistently preserve the low-rank constraint for memory efficiency, while achieving full-rank training (i.e., training with full-rank gradients of full-rank weights) to avoid inferior outcomes? In this paper, we propose a new plug-and-play training framework for LLMs called Fira, as the first attempt to achieve this goal. First, we observe an interesting phenomenon during LLM training: the scaling impact of adaptive optimizers (e.g., Adam) on the gradient norm remains similar from low-rank to full-rank training. Based on this observation, we propose a norm-based scaling method, which utilizes the scaling impact of low-rank optimizers as substitutes for that of original full-rank optimizers to enable full-rank training. In this way, we can preserve the low-rank constraint in the optimizer while achieving full-rank training for better performance. Moreover, we find that there are sudden gradient rises during the optimization process, potentially causing loss spikes. To address this, we further put forward a norm-growth limiter to smooth the gradient via regulating the relative increase of gradient norms. Extensive experiments on the pre-training and fine-tuning of LLMs show that Fira outperforms both LoRA and GaLore, achieving performance that is comparable to or even better than full-rank training. For instance, our Fira can reduce the memory usage of optimizer states by 61.1%, while achieving improved performance for pre-training on the LLaMA 1B architecture. Notably, for pre-training on the LLaMA 7B architecture, our method uses an $8\times$ smaller rank than GaLore, yet outperforms it by a large margin.

## 1 INTRODUCTION

In recent years, Large Language Models (LLMs) have achieved remarkable advancements in various domains (Achiam et al., 2023; Sima et al., 2023; Feng et al., 2024). While the substantial increase in model size contributes significantly to these advancements, it also introduces considerable memory bottlenecks, especially for optimizer states (Zhao et al., 2024a). For instance, pre-training a LLaMA-7B model from scratch [1] requires at least 58 GB memory, allocated as follows: 14GB for loading parameters, 14GB for weight gradients, 28GB for Adam (Kingma & Ba, 2014) optimizer states, and 2GB for activations (Zhao et al., 2024a). Notably, the optimizer states consume even more memory than the parameters themselves. To address this, low-rank training has demonstrated its effectiveness to reduce the memory usage of the optimizer states by conducting training in a low-rank subspace (Zhao et al., 2024a; Hu et al., 2022).

The current low-rank training methods can be broadly divided into two categories: weight matrix-based and gradient matrix-based low-rank decomposition. For the weight matrix decomposition methods, the most representative one is Low-Rank Adaptation (LoRA) (Hu et al., 2022), where its basic idea is to use low-rank matrices as decomposed representations of the pre-trained weights

---

[1] Training the model with a single batch size and maximum sequence length of 2048 under BF16 precision.

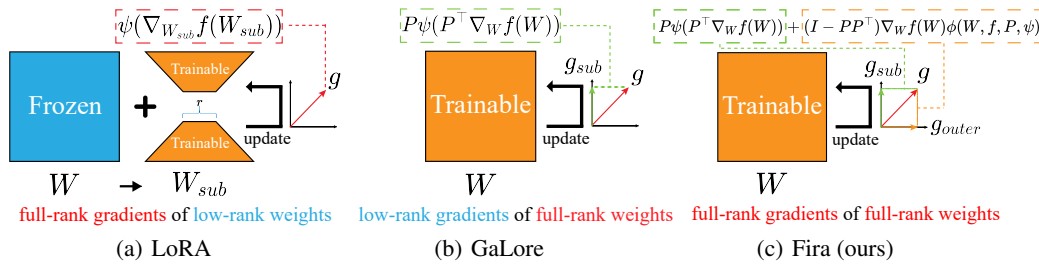

Figure 1: This analyses three types of memory-efficient approaches at a macro level.

during training, as shown in Figure 1 (a). However, the optimization of LoRA is constrained in a low-rank subspace of the weights. This will inevitably cause the reduction of representation capacity, leading to sub-optimal outcomes (Zhang et al., 2023b; Xia et al., 2024). Although the variant ReLoRA (Lialin et al., 2024) attempts to extend the application of LoRA from fine-tuning to pre-training, by periodically updating high-rank weights with multiple low-rank updates. It still requires full-rank weight training as a warm-up before low-rank training, thus rendering memory efficiency unachievable (Zhao et al., 2024a). For the gradient matrix decomposition based methods, the typical one is the gradient low-rank projection (GaLore) proposed recently (Zhao et al., 2024a). In contrast to LoRA, GaLore attempts to reduce the memory usage in optimizer states via decomposing the gradient matrix, as shown in Figure 1 (b). While GaLore supports the training of full-rank weights, it leverages only low-rank gradients, restricting them to a low-rank subspace. Consequently, any gradient information outside this subspace is lost, in contrast to training with full-rank gradients. Note that since these methods constrain LLM training to a low-rank subspace, this inevitably leads to sub-optimal results compared to full-rank training (i.e., training with full-rank gradients and full-rank weights). This raises the question: **Can we achieve full-rank training for LLMs while consistently maintaining a low-rank constraint?**

In light of this, we propose a new memory-efficient training framework for LLMs, called **Fira**, which, to the best of our knowledge, is the first to achieve **f**ull-rank tra**i**ning while consistently maintaining a low-**ra**nk constraint. To achieve this goal, a significant challenge is that the low-rank constraint makes it hard to preserve complete optimizer states (e.g., gradient momentum and variance) of full-rank weights in the commonly-used adaptive optimizer (e.g., Adam). As a result, the adaptive optimizer fails to correct the full-rank raw gradient according to the optimizer states. Without this correction, adaptive optimization algorithms would degrade into simple SGD, leading to significantly reduced optimization performance (Kingma & Ba, 2014; Zhang et al., 2020). This point is further validated in Section 4.1 and Section 5.4. Fortunately, we observe an interesting phenomenon during LLM training: the *scaling factor* [2] of the optimizer (e.g., Adam) is similar from low-rank training to full-rank training. As illustrated in Figure 3, if we sort the weight matrices by their average scaling factors, we can obtain a similar rank order. A detailed quantitative analysis of this similarity is presented in Appendix A.5 and A.6.

Based on this observation, we put forward a norm-based scaling method that utilizes the scaling factor of a weight matrix in low-rank training to replace the corresponding matrix's scaling factor in full-rank training. In this way, our scaling factor can also play a similar role in correcting the raw gradient, as adaptive optimizers do. Therefore, we can enable full-rank training while preserving the low-rank constraint. Additionally, we observe sudden increases in the gradient during training, which results in a spike in training loss (as depicted in Figure 4). This issue could lead to substantial parameter updates, causing the loss function to reach a much higher value and undermining prior optimization efforts (Goodfellow et al., 2016; Zhang et al., 2020). Despite the use of gradient clipping techniques (Pascanu et al., 2013), this issue may not be adequately resolved, as shown in Figure 4. To this end, we propose a norm-growth limiter, which aims to smooth the gradient by restricting the magnitude of the gradient norm's increase. By employing our limiter, we adaptively convert sudden rises in gradients into gradual increases, thereby facilitating a smooth update that mitigates the problem of loss spikes.

---

[2]The scaling factor $\phi_t(R_t)$ is defined as $\frac{||\psi(R_t)||}{||R_t||}$, where $||R_t||$ is the norm of the raw gradient, $||\psi(R_t)||$ is the norm of the gradient corrected by the gradient correction function $\psi$ of the optimizer (e.g., Adam).

Our main contributions can be summarized as follows:

1. We propose Fira, a plug-and-play memory-efficient training framework of LLMs, constituting the first attempt to enable full-rank training consistently under the low-rank constraint. We will release the source code and package of our Fira into a Python library for easy use.

2. We design two components in Fira: a norm-based scaling strategy that leverages the scaling effects of low-rank optimizers to facilitate full-rank training, and a norm-growth limiter to address the issue of loss spikes by limiting the growth of gradient norm.

3. Extensive experiments across various parameter counts (60M, 130M, 350M, 1B, 7B) validate the effectiveness of Fira in both pre-training and fine-tuning tasks. Our framework not only outperforms both LoRA and GaLore, but also achieves performance comparable to or better than full-rank training.

## 2 RELATED WORK

**Low-rank Adaptation.** Low-Rank Adaptation (LoRA) has been introduced by Hu et al. (2022) as an efficient fine-tuning method for LLMs. The core idea of LoRA is to freeze the pre-trained weights and introduce trainable low-rank matrices as decomposed representations of the pre-trained weights. In this way, the memory usage of training LLMs could be saved. Recently, a variety of methods by extending LoRA have been proposed to further improve the performance (Zhang et al., 2023c; Wen & Chaudhuri, 2023; Xia et al., 2024; Zhang et al., 2023b; Dettmers et al., 2024). For instance, ReLoRA (Lialin et al., 2024) is proposed to extend the application of LoRA from fine-tuning to pre-training. However, it still requires full-rank warm-up training before low-rank training, which prevents achieving memory efficiency. It is worth noting that while LoRA based methods reduce memory usage by limiting training to a low-rank parameter subspace, they inevitably reduce representation capacity (Xia et al., 2024).

**Gradient Projection.** Recent works (Zhang et al., 2023b; Xia et al., 2024; Valipour et al., 2022) have indicated that LoRA may yield sub-optimal performance since its low-rank constraints in parameters. Inspired by traditional projected gradient descent methods (Chen & Wainwright, 2015; Chen et al., 2019), GaLore (Zhao et al., 2024a) has been proposed recently to mitigate this problem. It enables full-parameter learning under low-rank constraints by projecting the gradient into a low-rank subspace, reducing memory usage for optimizer states. However, while GaLore allows memory-efficient full-parameter training, it confines the gradient to a low-rank subspace, discarding the portion outside the subspace and resulting in significant information loss. Inspired by Galore, several new memory-efficient methods have been developed. Flora (Hao et al., 2024) improves efficiency by randomly generating projection matrices as an alternative to the SVD method. Besides, LISA (Pan et al., 2024) enhances memory efficiency by freezing certain layers during optimization.

**System-Based Memory-Efficient Techniques.** Many system-based techniques have been developed to reduce memory usage in LLM training (Chen et al., 2016; Ren et al., 2021). However, most of these methods achieve memory efficiency by compromising either time or precision. Gradient checkpointing (Chen et al., 2016) is proposed to reduce memory usage by trading increased computational time for the re-computation of activations. Quantization (Dettmers et al., 2024) reduces memory consumption by using lower-bit data types, but at the cost of model precision. Memory offloading (Zhang et al., 2023a; Ren et al., 2021) reduces GPU memory usage by using non-GPU memory (e.g., CPU) as an extension. However, it introduces additional communication overhead, such as CPU-GPU transfer time. It's important to note that our proposed method is complementary to these approaches and can potentially be combined with them to further reduce memory usage.

## 3 PRELIMINARIES

### 3.1 REGULAR FULL-RANK TRAINING

At time step $t$, we denote the full-rank weight matrix as $W_t \in \mathbb{R}^{m \times n}$. The full-rank gradient can be represented as $G_t = \nabla_W f_t(W_t) \in \mathbb{R}^{m \times n}$, where $f$ is the objective function. Then the regular full-rank training can be expressed as follows:

$$W_{t+1} = W_t - \eta \psi_t(G_t), \tag{1}$$

where $\eta$ is the learning rate, and $\psi_t$ is the gradient correction function of the optimizer (for vanilla SGD, $\psi_t(G_t) = G_t$). Instead of vanilla SGD, adaptive optimizers (e.g., Adam (Kingma & Ba, 2014), AdamW (Loshchilov & Hutter, 2019)) are usually employed to correct the raw gradient for improving the training performance. However, this typically requires additional memory for storing optimizer states used in gradient correction. For instance, Adam (Kingma & Ba, 2014) requires storing the optimizer states $M$ and $V$, which consume $2mn$ of memory. The gradient correction process is as follows:

$$M_t = \beta_1 M_{t-1} + (1 - \beta_1) G_t, \tag{2}$$

$$V_t = \beta_2 V_{t-1} + (1 - \beta_2) G_t^2, \tag{3}$$

$$\psi_t(G_t) = \frac{\sqrt{1 - \beta_2^t}}{1 - \beta_1^t} \cdot \frac{M_t}{\sqrt{V_t} + \epsilon}, \tag{4}$$

where all matrix operations are element-wise. $\beta_1$ and $\beta_2$ are Adam's hyper-parameters, and $\epsilon$ is a small constant (e.g., $1 \times 10^{-8}$) used for numerical stability. Since this regular full-rank training typically consumes a large amount of memory for training LLMs, many representative low-rank training methods, e.g., LoRA (Hu et al., 2022) and Galore (Zhao et al., 2024a), have been proposed to reduce memory usage in recent years.

### 3.2 LOW-RANK ADAPTATION

The basic idea behind LoRA (Hu et al., 2022) is to use low-rank matrices as decomposed representations of the pre-trained weights during training, in order to reduce memory usage. Formally, LoRA freezes the full-rank weight matrix $W_0 \in \mathbb{R}^{m \times n}$ and incorporates two low-rank matrices $A_t$ and $B_t$ for training as:

$$W_t = W_0 + B_t A_t, \tag{5}$$

where $B_t \in \mathbb{R}^{m \times r}$, $A_t \in \mathbb{R}^{r \times n}$, and the rank $r < \min(m, n)$. While LoRA reduces memory usage by limiting training to a low-rank subspace of the weight, it inevitably diminishes the representation capacity of the weight matrix $W_t$.

### 3.3 GRADIENT LOW-RANK PROJECTION

In contrast to LoRA, GaLore (Zhao et al., 2024a) utilizes a projection matrix $P_t \in \mathbb{R}^{m \times r}$ to project the full-rank gradient $G_t \in \mathbb{R}^{m \times n}$ to a low-rank gradient $R_t = P_t^\top G_t \in \mathbb{R}^{r \times n}$ $(m \leq n)^3$. By doing so, the memory usage of optimizer states could be reduced. The parameter update in GaLore can be formulated as:

$$W_{t+1} = W_t - \eta P_t \psi_t(R_t), \tag{6}$$

where the projection matrix $P_t$ can be obtained through singular value decomposition (SVD) of $G_t$ and can be updated every $T$ steps:

$$G_t = U \Sigma V^\top \approx \sum_{i=1}^{r} \sigma_i u_i v_i^\top, \quad P_t = [u_1, u_2, \ldots, u_r], \tag{7}$$

where $u_i$ is the $i$-th column vector of the left singular matrix $U$. By selecting the first $r$ columns of matrix $U$ that correspond to the largest singular values, the projection matrix $P_t$ effectively captures the most significant directions in the gradient space, leading to faster convergence (Zhao et al., 2024a). The optimal switching frequency $T$ is usually set to be between 50 to 1000, and the additional computational overhead introduced by SVD is negligible ($< 10\%$), as stated in (Zhao et al., 2024a). Since Galore restricts the gradient in the low-rank subspace, the gradient information outside this subspace is lost, leading to inferior performance.

## 4 PROPOSED METHOD

To achieve full-rank training under low-rank constraints, our framework, named Fira, consists of two important components: (i) a norm-based scaling method, enabling full-rank training by leveraging the scaling effects of adaptive optimizers; (ii) a norm-growth limiter, which restricts the growth of the gradient norm to prevent spikes in training loss. Next, we will elaborate on these two components.

---

[3]For simplicity, we assume $m \leq n$, following (Zhao et al., 2024a). If $m > n$, $R_t = G_t Q_t \in \mathbb{R}^{m \times r}$, $Q_t \in \mathbb{R}^{n \times r}$.

## 4.1 NORM-BASED SCALING

The low-rank constraint makes it challenging to record complete optimizer states for correcting raw gradients in full-rank training. Fortunately, we find an interesting phenomenon in LLM training: the scaling factor at the matrix level remains similar from low-rank training to full-rank training. Based on this observation, we propose a norm-based scaling strategy that approximately corrects the raw gradient, similar to adaptive optimizers, thereby enabling full-rank training.

**Challenge Analysis.** Given the difficulty of incorporating trainable low-rank weights into LoRA to achieve full-rank weight training (Zhao et al., 2024a), we focus on investigating how to achieve full-rank gradient training by extending the gradient projection method, Galore, in this paper. In GaLore, the projection matrix $P_t \in \mathbb{R}^{m \times r}$ projects the full-rank gradient $G_t \in \mathbb{R}^{m \times n}$ of the full-rank weight $W_t \in \mathbb{R}^{m \times n}$, to the low-rank subspace gradient $R_t = P_t^\top G_t \in \mathbb{R}^{r \times n}$. The gradient outside this subspace can be represented as: $(I - P_t P_t^\top)G_t = G_t - P_t R_t$. In other words, the full-rank gradient $G_t$ can be divided into two terms: $P_t R_t$ and $(G_t - P_t R_t)$.

In GaLore, the optimizer states only store the information of $R_t$ instead of $G_t$ to realize the low-rank constraint. The term of $(G_t - P_t R_t)$ is directly discarded in Galore due to the lack of corresponding optimizer states of $G_t$ for correction in optimizers. This would lead to significant information loss especially when $r \ll d_{model}$, where $d_{model} = \min(m, n)$ is the full-rank dimension of models (This point can be ver-

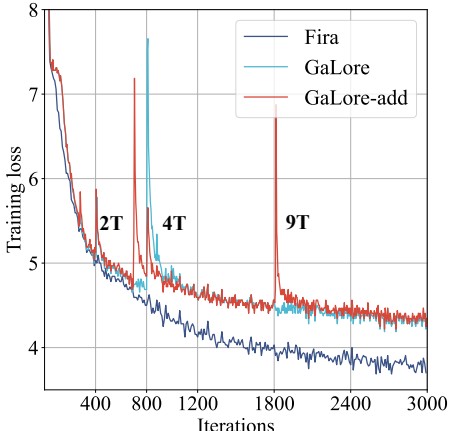

Figure 2: Training loss of different methods for pre-training LLaMA 60M on C4 dataset with $r/d_{model}$ = 16/256 and $T$ = 200.

ified in our experiment section, as illustrated in Figure 6. In Figure 6, the validation perplexity of GaLore significantly increases at $r = 4$ compared to $r = 128$ when $d_{model} = 256$, indicating a substantial loss of information and decreased training performance). Intuitively, to capture the information of $(G_t - P_t R_t)$, we can directly add it based on Eq. (6) as follows:

$$W_{t+1} = W_t - \eta P_t \psi_t(R_t) - \eta(G_t - P_t R_t). \tag{8}$$

We denote the update strategy in Eq. (8) as GaLore-add. However, as illustrated in Figure 2, GaLore-add exhibits almost no improvement compared to updates using Eq. (6) in GaLore. This phenomenon primarily arises because the term of $(G_t - P_t R_t)$ doesn't have corresponding optimizer states for gradient correction. As a result, the optimization of $(G_t - P_t R_t)$ uses vanilla SGD, yielding sub-optimal outputs. Besides, in GaLore-add, $P_t \psi_t(R_t)$ employs the Adam optimizer for training while $(G_t - P_t R_t)$ employs vanilla SGD. This gradient misalignment may also account for the lack of noticeable improvement.

**Similarity of Scaling Factor.** To tackle this challenge, we propose the concept of the *scaling factor*, which is defined as follows:

$$\phi_t(R_t) = \frac{||\psi_t(R_t)||}{||R_t||}, \tag{9}$$

where the scaling factor $\phi_t$ represents the magnitude of the correction applied by the adaptive optimizer to the gradient norm. Based on the scaling factor $\phi_t$, we observe an interesting phenomenon during LLM training: the scaling factors at the matrix level exhibit a high degree of similarity between low-rank and full-rank training. As shown in Figure 3, sorting weight matrices by their average scaling factors results in an almost similar order.

Based on this observation, we can use the scaling factors in low-rank training to replace those in full-rank training, even though the absolute magnitude of the scaling factors may vary between low-rank and full-rank training. This is because the absolute magnitude can be regarded as the learning rate, to which adaptive optimizers (e.g., Adam) are not sensitive (Zhao et al., 2024b; Liu et al., 2019). Instead, the discrepancies (or relative orders) in the correction magnitudes of different parameters (or weight matrices) are of greater significance.

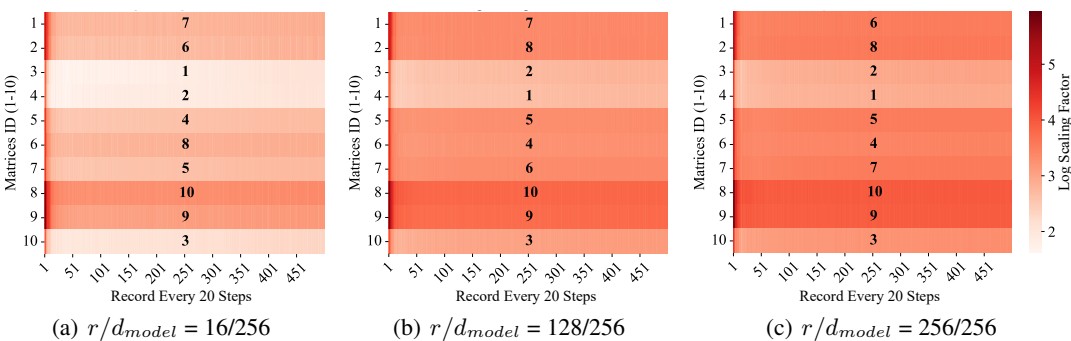

(a) $r/d_{model} = 16/256$    (b) $r/d_{model} = 128/256$    (c) $r/d_{model} = 256/256$

Figure 3: Scaling factor $\phi_t(R_t)$ of different weight matrices for pre-training LLaMA 60M on the C4 dataset for 10K steps with varying ranks. Here $x$-axis is the number of recorded steps and $y$-axis is the ID of the weight matrices. Each row is labeled according to the order of its average scaling factor value. Case study and lager-scale quantitative analysis are provided in Appendix A.5 and Appendix A.6, respectively.

**Norm-based Scaling.** Inspired by this, we propose a norm-based scaling method that utilizes the scaling factor of a weight matrix in low-rank training as a substitute for the corresponding factor in full-rank training:

$$W_{t+1} = W_t - \eta P_t \psi_t(R_t) - \eta \phi_t(R_t)(G_t - P_t R_t). \tag{10}$$

By Eq. (10), we can approximately correct $(G_t - P_t R_t)$ as adaptive optimizers do, so as to achieve full-rank training under low-rank constraints.

Furthermore, we can use a more fine-grained average of the scaling factor in Eq.(10), by considering each column in the weight matrix:

$$\phi_t(R_t)_i = \frac{||\psi(R_{t,:,i})||}{||R_{t,:,i}||}, \quad i = 1, 2, \ldots, n, \tag{11}$$

where $R_{t,:,i}$ is the $i$-th column of $R_t$, and $\phi_t(R_t)_i$ is the $i$-th scaling factor.

## 4.2 NORM-GROWTH LIMITER

We find that there are suddenly sharp increases of the gradient during training, which could introduce loss spikes. As shown in Figure 4, Fira-w.o.-limiter (our method without using the proposed norm-growth limiter) experiences spikes in both gradient norm and training loss. In this section, we analyze the reasons for this issue and propose a norm-growth limiter which transforms abrupt gradient spikes into gradual, smooth increases.

**Loss Spike Analysis.** There are two main reasons for the spikes: (i) Switching the projection matrix $P_t$ in gradient projection methods would cause instability during training. As illustrated in Figure 2, both GaLore and GaLore-add exhibit significant training loss spikes at integer multiples of $T$ (i.e., the frequency of switching the projection matrix $P_t$). This instability occurs because, when switching projection matrices $P_t$, the optimizer retains states linked to the previous matrix, while the current input gradient uses a new projection matrix, leading to significant misalignment. Furthermore, as shown in Figure 2, GaLore-add also exhibits training spikes, reinforcing our earlier claim that directly incorporating $(G_t - P_t R_t)$ may introduce instability and hinder training; (ii) Maintaining the original direction of the raw gradient $(G_t - P_t R_t)$ may be insufficient for handling the sharp loss landscapes in LLM training, unlike Adam (Zhang et al., 2020). Due to space constraints, further analysis is provided in Appendix A.7.

**Addressing Loss Spikes.** To address this issue, a straightforward solution is to use gradient clipping techniques (Pascanu et al., 2013) to avoid loss spikes. However, clipping based on the absolute norm of gradient matrices fails to account for significant differences between them, leading to sub-optimal results. This point can be also verified in Figure 4 and Table 4. To this end, we propose a norm-growth limiter method that constrains the ratio of the current gradient norm to the previous step's

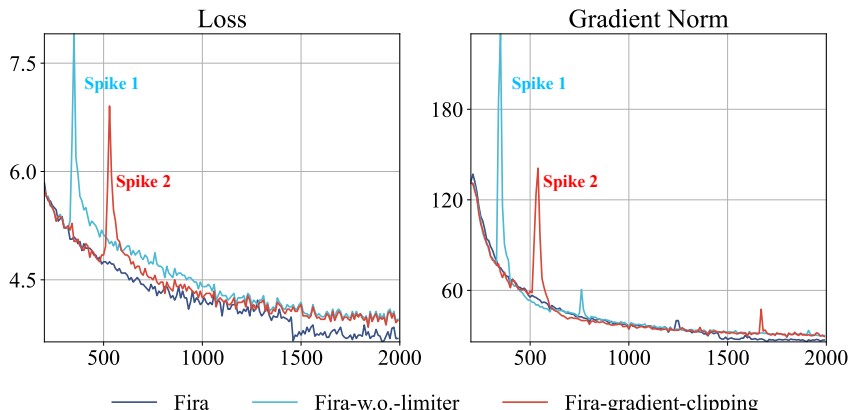

Figure 4: Training loss and gradient norm of three variants of Fira for pre-training LLaMA 60M on C4 dataset.

norm to a fixed ratio $\gamma$ when the gradient norm increases:

$$\text{if } \frac{||S_t||}{||S_{t-1}||} > \gamma \text{ then } S_t \leftarrow \frac{S_t}{||S_t||} \cdot \gamma ||S_{t-1}||, \tag{12}$$

where $\gamma$ is a threshold ensuring that the rate of gradient growth does not exceed this value. $S_t = \phi_t(R_t)(G_t - P_t R_t)$ is the corrected gradient by applying our norm-based scaling. This approach limits the magnitude of gradient norm increases, converting sudden spikes into gradual rises and thus preventing loss spikes. Moreover, by constraining the relative increase of each gradient matrix's norm, our method is more flexible than the absolute norm clipping. As illustrated in Figure 2 and Figure 4, Fira with our proposed limiter improves the optimization performance without significant spikes.

---

**Algorithm 1** Fira with Adam

**Input:** $\eta$ : step size, $\{\beta_1, \beta_2\}$ : decay rates, $W \in \mathbb{R}^{m \times n}$ with $m \leq n$ : weight matrices, $r$ : rank, $T$ : switching frequency, $\alpha$ : hyper-parameter of Galore, $\gamma$ : limiter threshold.
**Output:** $W_t$ : resulting weight matrix
1: $M_0, V_0 \in \mathbb{R}^{r \times n} \leftarrow 0, 0 \quad t \leftarrow 0$         ▷ Initialize moving 1st, 2nd moment and step
2: **repeat**
3:     $G_t \in \mathbb{R}^{m \times n} \leftarrow \nabla_W f_t(W_t)$         ▷ Calculate full-rank gradients of full-rank weights
4:     **if** $t \bmod T = 0$ **then**
5:        $U, \Sigma, V^\top \leftarrow \text{SVD}(G_t) \quad P_t \leftarrow U[:, :r]$ ▷ Initialize the projection matrix every $T$ steps
6:     **else**    $P_t \leftarrow P_{t-1}$         ▷ Reuse the previous projection matrix
7:     **end if**
8:     $R_t, S_t \leftarrow P_t^\top G_t, (I - P_t P_t^\top)G_t$ ▷ Divide gradients into two terms by gradient projection

---

9:     $M_t \leftarrow \beta_1 M_{t-1} + (1 - \beta_1)R_t$         ▷ $\psi_t(R_t)$: Apply Adam with low-rank gradients $R_t$
10:    $V_t \leftarrow \beta_2 V_{t-1} + (1 - \beta_2)R_t^2$
11:    $N_t \leftarrow \frac{\sqrt{1-\beta_2^t}}{1-\beta_1^t} \cdot \frac{M_t}{\sqrt{V_t}+\epsilon}$

---

12:    $K \leftarrow [\frac{||N_t[:, 1]||}{||R_t[:, 1]|| + \epsilon}, \frac{||N_t[:, 2]||}{||R_t[:, 2]|| + \epsilon}, \cdots, \frac{||N_t[:, n]||}{||R_t[:, n]|| + \epsilon}]$         ▷ Norm-Based Scaling
13:    $S_t \leftarrow [k_1 S_t[:, 1], k_2 S_t[:, 2], \cdots, k_n S_t[:, n]]$
14:    $S_t \leftarrow S_t \cdot \gamma / \max\{\frac{||S_t||}{||S_{t-1}|| + \epsilon}, \gamma\}$         ▷ Norm-Growth Limiter

---

15:    $\tilde{G}_t \leftarrow \alpha \cdot (P_t N_t + S_t)$         ▷ Project back and complete full-rank gradients
16:    $W_t \leftarrow W_{t-1} - \eta \cdot \tilde{G}_t \quad t \leftarrow t + 1$         ▷ Update the weight matrix
17: **until** convergence criteria met
18: **return** $W_T$

---

Table 2: Comparison of different methods for pre-training LLaMA models of various sizes on the C4 dataset. We report validation perplexity ($\downarrow$) with a memory estimate of total parameters and optimizer states. Results and memory estimates of all baselines are taken from Zhao et al. (2024a). $r$ refers to the rank and $d_{model}$ is the full-rank dimension of models.

|  | **60M** | **130M** | **350M** | **1B** |
|---|---|---|---|---|
| Full-Rank | 34.06 (0.36G) | 25.08 (0.76G) | 18.80 (2.06G) | 15.56 (7.80G) |
| **Fira** | **31.06** (0.24G) | **22.73** (0.52G) | **16.85** (1.22G) | **14.31** (4.38G) |
| GaLore | 34.88 (0.24G) | 25.36 (0.52G) | 18.95 (1.22G) | 15.64 (4.38G) |
| LoRA | 34.99 (0.36G) | 33.92 (0.80G) | 25.58 (1.76G) | 19.21 (6.17G) |
| ReLoRA | 37.04 (0.36G) | 29.37 (0.80G) | 29.08 (1.76G) | 18.33 (6.17G) |
| $r / d_{model}$ | 128 / 256 | 256 / 768 | 256 / 1024 | 512 / 2048 |
| Training Tokens | 1.1B | 2.2B | 6.4B | 13.1B |

## 4.3 OVERALL ALGORITHM

We present the overall algorithm of Fira with Adam in Algorithm 1. Our main components, the norm-based scaling method and the norm-growth limiter, are straightforward to implement, requiring only 3 additional lines of code. Moreover, Fira is a plug-and-play framework which can be easily integrated into the training process without requiring significant modifications. The plug-and-play Pytorch-like pseudo-code of Fira is provided in Appendix A.4.

Table 1: Comparison between Fira, GaLore, and LoRA. Denote $W_t \in \mathbb{R}^{m \times n}$ ($m \leq n$), rank $r$.

|  | Fira | GaLore | LoRA |
|---|---|---|---|
| Weights | $mn$ | $mn$ | $mn + mr + nr$ |
| Optimizer States | $mr + 2nr + 1$ | $mr + 2nr$ | $2mr + 2nr$ |
| Full-Rank Gradients | ✓ | ✗ | ✓ |
| Full-Rank Weights | ✓ | ✓ | ✗ |
| Pre-Training | ✓ | ✓ | ✗ |
| Fine-Tuning | ✓ | ✓ | ✓ |

It's worth noting that Fira only introduces one parameter $||S_{t-1}||$ for each weight matrix in the optimizer state, which is negligible, as shown in Table 1. Besides, in addition to the original hyper-parameters of optimizers and gradient projection methods, Fira only adds one hyper-parameter $\gamma$ in the norm-growth limiter. The hyper-parameter $\gamma$ is set to 1.01 across all experiments, which consistently yields satisfactory results.

## 5 EXPERIMENTS

In this section, we validate the effectiveness of Fira in pre-training and fine-tuning tasks of LLMs. In our experiments, we denote our method using the strategy of Eq. (10) as Fira-matrix, and denote our method additionally using the column-wise strategy of Eq. (11) as Fira.

### 5.1 MEMORY-EFFICIENT PRE-TRAINING

**Experimental Setup.** We follow the settings in Galore (Zhao et al., 2024a) to conduct the pre-training experiments. We compare Fira with GaLore (Zhao et al., 2024a), LoRA (Hu et al., 2022), ReLoRA (Lialin et al., 2024), and full-rank training baselines. Adam optimizer is used for training all baselines and our method on the C4 dataset in the BF16 format. The settings of these baselines can be found in Zhao et al. (2024a). The dataset C4 is a colossal, cleaned version of Common Crawl's web crawl corpus, which is widely used in LLM pre-training (Raffel et al., 2020). Following Zhao et al. (2024a), we utilize LLaMA-based architectures equipped with RMSNorm and SwiGLU activations (Zhang & Sennrich, 2019; Shazeer, 2020; Touvron et al., 2023). As in Zhao et al. (2024a), our training protocol excludes data repetition and spans a sufficiently large dataset, encompassing a diverse array of model sizes (60M, 130M, 350M, 1B). To guarantee a fair comparison, we employ the same learning rate 0.01 as used in GaLore and maintain the same rank $r$ for each model size. The detailed settings of pre-training are provided in Appendix A.2. We use 8 A100 80G GPUs to conduct pre-training experiments.

**Result Analysis.** As shown in Table 2, Fira consistently outperforms low-rank training baselines by a large margin under the same rank constraint, and even surpasses full-rank training. Following Zhao et al. (2024a), we estimate the memory reduction of the optimizer states via the same memory estimation method introduced in Zhao et al. (2024a). From Table 2, our Fira saves 61.1% memory usage of the optimizer states when pre-training the LLaMA 1B architecture compared to full-rank training, while Fira achieving better results. Compared to full-rank training, Fira's superior performance may be attributed to the following reason: the gradient direction in the norm-based scaling method is determined by the current state, rather than by historical gradients in Adam. Therefore, Fira introduces a higher degree of randomness in training, which can enhance the model's ability to escape the local optima, leading to better training performance (Zhou et al., 2020).

### 5.2 SCALING UP TO LLaMA 7B PRE-TRAINING.

To validate the scalability of our method, we scale up by pre-training the LLaMA 7B model with the full-rank dimension $d_{\text{model}} = 4096$. We compare Fira with the GaLore baseline, which generally achieves the best performance among low-rank training baselines, as shown in Table 2. As illustrated in Figure 5, our method demonstrates a significant improvement over GaLore for pre-training LLaMA 7B, while using an $8\times$ smaller rank. This highlights Fira's effectiveness, suggesting it could be a viable solution for large-scale LLM pre-training.

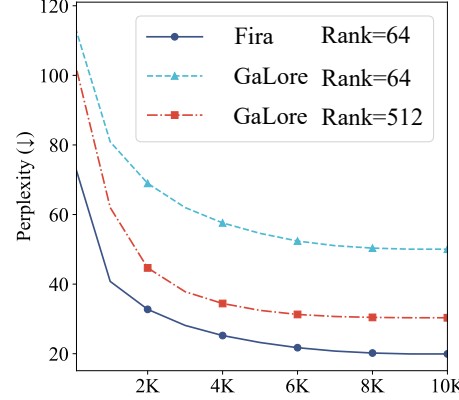

Figure 5: Pre-training LLaMA 7B with different methods on the C4 dataset.

### 5.3 MEMORY-EFFICIENT FINE-TUNING

**Experimental Setup.** Following Hu et al. (2023), we perform the fine-tuning task to compare Fira with LoRA, GaLore, Flora, ReLoRA, Full-rank training, and other baseline methods, including Prefix-tuning (Prefix) (Li & Liang, 2021), Series Adapter (Series) (Houlsby et al., 2019), and Parallel Adapter (Parallel) (He et al., 2021), on the LLaMA-7B model for commonsense reasoning tasks. This task consists of eight sub-tasks, each with its own designated training and testing sets. Following the approach of Hu et al. (2023), we combine the training datasets from all eight sub-tasks into a unified training set, while evaluating each sub-task individually using its respective testing dataset. In the fine-tuning task, the rank $r$ is set to 32 and the learning rate is set to 1e-4. The detailed settings of fine-tuning are provided in Appendix A.3. We adopt RTX 4090 GPUs for fine-tuning experiments.

**Result Analysis.** As shown in Table 3, our Fira achieves the highest performance on 4 out of 8 datasets, demonstrating better or comparable performance compared to the baseline methods. Notably, GaLore struggles to adapt to the HellaSwag and WinoGrande datasets, resulting in a significant decline in scores. In contrast, our Fira adapts to these tasks well and achieves the highest scores on WinoGrande. In terms of memory efficiency, our method uses comparable or even less memory than the low-rank training methods LoRA and GaLore. These results illustrate the effectiveness of our method for the fine-tuning of LLMs.

Table 3: Accuracy ($\uparrow$) of various fine-tuning methods on eight commonsense reasoning datasets with LLaMA 7B. Results for all baseline methods, except GaLore, are taken from Hu et al. (2023).

| Method | Memory | BoolQ | PIQA | SIQA | HellaSwag | WinoGrande | ARC-e | ARC-c | OBQA | Avg |
|---|---|---|---|---|---|---|---|---|---|---|
| **Fira** | 14.26G | 69.4 | **82.6** | 78.0 | 76.8 | **81.2** | **82.2** | 64.4 | **80.8** | **76.9** |
| Prefix | 14.05G | 64.3 | 76.8 | 73.9 | 42.1 | 72.1 | 72.9 | 54.0 | 60.6 | 64.6 |
| Series | 14.42G | 63.0 | 79.2 | 76.3 | 67.9 | 75.7 | 74.5 | 57.1 | 72.4 | 70.8 |
| Parallel | 15.49G | 67.9 | 76.4 | **78.8** | 69.8 | 78.9 | 73.7 | 57.3 | 75.2 | 72.2 |
| LoRA | 14.35G | 68.9 | 80.7 | 77.4 | **78.1** | 78.8 | 77.8 | 61.3 | 74.8 | 74.7 |
| ReLoRA | 14.35G | 68.9 | 81.2 | 77.8 | 46.0 | 79.4 | 80.2 | 64.2 | 79.6 | 72.2 |
| Flora | 14.26G | 50.1 | 77.5 | 74.2 | 53.8 | 45.5 | 79 | 64.6 | 74.8 | 64.9 |
| GaLore | 14.26G | **69.5** | 82.0 | 75.1 | 32.2 | 18.0 | 80.7 | **65.8** | 78.0 | 62.7 |
| Full-rank | 42.00G | 64.2 | 68.1 | 68.0 | 42.3 | 66.5 | 55.6 | 43.9 | 60.0 | 58.6 |

### 5.4 ABLATION STUDY

In this section, we conduct an ablation study to assess the effectiveness of each component in our method. We adopt the same settings in Section 5.1 for pre-training the LLaMA 60M model. We design four variants of our method for the ablation study: (1) **Fira-w.o.-scaling:** our Fira without using the scaling factor to correct the gradient (i.e., setting $\phi_t(R_t)$ to a fixed value of 1). (2) **Fira-matrix:** our Fira using the scaling factor at the matrix level instead of at the column level. (3) **Fira-w.o.-limiter:** our Fira without using norm-growth limiter to avoid training loss spikes. (4) **Fira-gradient-clipping:** our Fira using gradient clipping to avoid loss spikes instead of our proposed norm-growth limiter.

Table 4 presents the results. It can be found that Fira outperforms Fira-w.o.-scaling, thereby demonstrating the effectiveness of our proposed norm-based scaling method for gradient correction. This also suggests that directly incorporating the raw gradient outside the subspace without correction will lead to sub-optimal results. Besides, Fira yields better performance than Fira-matrix, illustrating that a more fine-grained consideration of the scaling factor is beneficial. Furthermore, Fira demonstrates improved performance over Fira-w.o.-limiter and Fira-gradient-clipping, indicating the effectiveness of our proposed norm-growth limiter in addressing the issue of training loss spikes.

Table 4: Ablation study on the C4 dataset.

| Method | Perplexity ($\downarrow$) |
|---|---|
| Fira-w.o.-scaling | 37.06 |
| Fira-matrix | 31.52 |
| Fira-w.o.-limiter | 32.22 |
| Fira-gradient-clipping | 31.22 |
| Fira | **31.06** |

### 5.5 PERFORMANCE UNDER VARYING RANKS

In this section, we illustrate the advantages of our Fira over Galore under a lower rank. We adjust various rank configurations within the set $\{4, 16, 64, 128\}$ and $d_{model} = 256$, and then assess the performance of pre-training the LLaMA 60M model on the C4 dataset as outlined in Section 5.1. The validation perplexity of Fira and GaLore after 10K steps across different ranks is depicted in Figure 6. From Figure 6, we can observe that Fira consistently surpasses GaLore across all rank configurations. Notably, even when the ranks are set very low (4 and 16), Fira still achieves performance comparable to full-rank training. In contrast, the performance of GaLore significantly declines in these cases. These results highlight the superiority of our proposed Fira at lower ranks and its effectiveness in reducing memory usage.

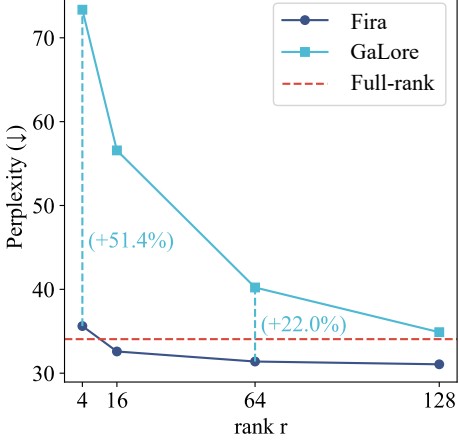

Figure 6: Validation perplexity of Fira and GaLore for varying ranks when pre-training LLaMA 60M on the C4 dataset with $d_{model} = 256$.

## 6 CONCLUSION

In this paper, we present a plug-and-play memory-efficient training framework for LLMs, called Fira, as the first attempt to facilitate full-rank training consistently under low-rank constraints. First, we find a notable phenomenon in LLM training: the scaling effect of adaptive optimizers on the gradient norm remains similar between low-rank and full-rank training. Building on this observation, we propose a norm-based scaling method that applies the scaling effect of low-rank optimizers in place of full-rank optimizers to facilitate full-rank training. This allows us to maintain the low-rank constraint within the optimizer while still benefiting from the advantages of full-rank training for improved performance. Additionally, we observe there are sudden spikes in gradient values during optimization, which could lead to spikes in loss. To mitigate this, we propose a norm-growth limiter that smooths gradients by regulating the relative increase in gradient norms. Extensive experiments in both pre-training and fine-tuning of LLMs demonstrate the effectiveness of our proposed Fira.

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

# A  APPENDIX

## A.1  THEORETICAL ANALYSIS

**Error Upper Bound for the Approximation of Scaling Factors.** In Section 4.1, we use the scaling factors of low-rank gradients to approximate that of full-rank gradients. To quantify the effectiveness of this approximation, we will derive its error upper bound theoretically, and verify our analysis experimentally. The error of the approximation $\kappa(r)$ can be written as:

$$\kappa(r) = \left| \phi_t^2(G_t) - \phi_t^2(R_t) \right|, \tag{13}$$

where rank $r \leq n$, and $R_t = P_t^\top G_t \in \mathbb{R}^{r \times n}$. To simplify the proof, we consider that $r$ components $(g_1, \ldots, g_r)$ of low-rank gradients are directly sampled from $n$ components $(g_1, \ldots, g_n)$ of full-rank gradients. Under these conditions, the error can be rewritten as:

$$\kappa(r) = \left| \frac{\sum_{i=1}^n \psi_i^2(g_i)}{\sum_{i=1}^n g_i^2} - \frac{\sum_{i=1}^r \psi_i^2(g_i)}{\sum_{i=1}^r g_i^2} \right|. \tag{14}$$

**Assumption 1.** (Bounded Scaling Factors): we assume that the adaptive optimizer scales each gradient component $g_i$ by the scaling factor that lies within known bounds. Specifically, there exist constants $c_{\min}$ and $c_{\max}$ such that for all $i$:

$$c_{\min} \leq \left| \frac{\psi_i(g_i)}{g_i} \right| \leq c_{\max}. \tag{15}$$

This implies:

$$c_{\min}^2 \leq \frac{\psi_i^2(g_i)}{g_i^2} \leq c_{\max}^2. \tag{16}$$

**Theorem 1.** (Error Upper Bound for Approximation) Under the assumption that $\frac{\psi_i^2(g_i)}{g_i^2}$ are bounded between constants $c_{\min}^2$ and $c_{\max}^2$ for all components $i$, the approximation error $\kappa(r)$ satisfies:

$$\kappa(r) \leq (c_{\max}^2 - c_{\min}^2) \cdot (1 - \frac{\sum_{i=1}^r g_i^2}{\sum_{i=1}^n g_i^2}). \tag{17}$$

*Proof.* We first define the following quantities: Total Gradient Norm $s_n = \sum_{i=1}^n g_i^2$, Partial Gradient Norm (first $r$ components) $s_r = \sum_{i=1}^r g_i^2$, Remaining Gradient Norm $s_{n-r} = s_n - s_r = \sum_{i=r+1}^n g_i^2$, Total Corrected Gradient Norm $S_n = \sum_{i=1}^n \psi_i^2(g_i)$, Partial Adjusted Gradient Norm (first $r$ components) $S_r = \sum_{i=1}^r \psi_i^2(g_i)$, Remaining Adjusted Gradient Norm $S_{n-r} = S_n - S_r = \sum_{i=r+1}^n \psi_i^2(g_i)$.

Then, our estimation error $\kappa(r)$ can be rewritten using these definitions:

$$\kappa(r) = \left| \frac{S_n}{s_n} - \frac{S_r}{s_r} \right|. \tag{18}$$

First, we rewrite the estimation error $\kappa(r)$ in terms of $S_r$, $S_{n-r}$, $s_r$, and $s_{n-r}$:

$$\kappa(r) = \left| \frac{S_r + S_{n-r}}{s_r + s_{n-r}} - \frac{S_r}{s_r} \right|. \tag{19}$$

Then, compute the difference in the numerator:

$$\kappa(r) = \left| \frac{(S_r + S_{n-r})s_r - S_r(s_r + s_{n-r})}{(s_r + s_{n-r})s_r} \right|. \tag{20}$$

Simplify the numerator, thus, the estimation error becomes:

$$\kappa(r) = \frac{|S_{n-r}s_r - S_r s_{n-r}|}{s_n s_r}. \tag{21}$$

After that, we factor out $s_{n-r}$:

$$\kappa(r) = \frac{s_{n-r}}{s_n} \cdot \left| \frac{S_{n-r}}{s_{n-r}} - \frac{S_r}{s_r} \right|. \tag{22}$$

From our bounded assumption, we have:

$$c_{\min}^2 \leq \frac{S_r}{s_r} \leq c_{\max}^2 \quad \text{and} \quad c_{\min}^2 \leq \frac{S_{n-r}}{s_{n-r}} \leq c_{\max}^2. \tag{23}$$

Therefore, the maximum possible difference between $\frac{S_{n-r}}{s_{n-r}}$ and $\frac{S_r}{s_r}$ is:

$$\left| \frac{S_{n-r}}{s_{n-r}} - \frac{S_r}{s_r} \right| \leq c_{\max}^2 - c_{\min}^2. \tag{24}$$

Finally, since $\frac{s_{n-r}}{s_n} = (1 - \frac{\sum_{i=1}^{r} g_i^2}{\sum_{i=1}^{n} g_i^2})$, the approximation error $\kappa(r)$ is bounded above by:

$$\kappa(r) \leq (c_{\max}^2 - c_{\min}^2) \cdot (1 - \frac{\sum_{i=1}^{r} g_i^2}{\sum_{i=1}^{n} g_i^2}). \tag{25}$$

$\square$

From this theory, we can find that the error upper bound on the approximation of scaling factors is mainly determined by two aspects, and we can verify them experimentally:

- Variability of Scaling Factor $(c_{\max}^2 - c_{\min}^2)$: This term represents the maximum variation in the scaling factors of different gradient components. For further validation, we designed *Fira-only-scaling*, a variant of Fira. It directly applies the low-rank scaling factors to the full-rank gradients by changing the Eq. (10) from $W_{t+1} = W_t - \eta P_t \psi_t(R_t) - \eta \phi_t(R_t)(G_t - P_t R_t)$ to $W_{t+1} = W_t - \eta \phi_t(R_t) G_t$. In this way, we are able to exclude the influence of the original Adam term $P_t \psi_t(R_t)$ and better analyze the effectiveness of our approximation. As shown in Table 5, Fira-only-scaling (column-level) gains better performance than Fira-only-scaling (matrix-level) for its more fine-grained consideration of the scaling factor, which also means a smaller maximum variation $(c_{\max}^2 - c_{\min}^2)$.

- Effectiveness of Gradient Sampling $(1 - \frac{\sum_{i=1}^{r} g_i^2}{\sum_{i=1}^{n} g_i^2})$: This term represents the proportion of the gradients norm contributed by the sampled low-rank $r$ components from full-rank $n$ components. As shown in Table 6, we conducted ablation experiments *Fira-only-scaling-w.o.-svd*, i.e., Fira-only-scaling without SVD in low-rank gradient sampling. As we can see, SVD is capable of sampling more prominent low-rank gradients, which leads to a reduction in the upper bound of error and enhanced performance. Similarly, as shown in Table 7, employing a higher rank enables the sampling of a greater proportion of the gradients norm, resulting in reducing error upper bound and improved performance.

Table 5: Ablation on the level of scaling factors for the variant Fira-only-scaling.

| Level | Perplexity ($\downarrow$) |
|---|---|
| Column | 31.68 |
| Matrix | 32.05 |

Table 6: Ablation on SVD for the variant Fira-only-scaling.

| Method | Perplexity ($\downarrow$) |
|---|---|
| Fira-only-scaling | 31.68 |
| Fira-only-scaling-w.o.-svd | 32.22 |

Table 7: Ablation on rank for the variant Fira-only-scaling.

| Rank | 4 | 16 | 64 | 128 |
|---|---|---|---|---|
| Perplexity ($\downarrow$) | 35.91 | 32.90 | 31.93 | 31.68 |

**Variance of Scaling Factors.** The variance of adaptive learning rates is significantly elevated during the early stage of training, often necessitating a warm-up to mitigate this variance and stabilize training (Liu et al., 2019). As illustrated in Figure 7, the scaling factor in Fira exhibits a similar pattern, characterized by substantial variance during the early stage of training, which also necessitates a warm-up. However, the addition of an extra warm-up hyper-parameter for Fira would be inefficient. Therefore, it is crucial to investigate whether the original warm-up would have mitigated the variance in Fira efficiently. In the subsequent theoretical analysis, we show that, during the early training phase, the variance of the scaling factor of Fira is less than or equal to that of the adaptive learning rate. This finding suggests that the existing warm-up strategy is sufficient to mitigate the variance of Fira, thereby eliminating the need for an additional warm-up hyper-parameter.

Consider independent random vectors $\{g^{(i)}\}_{i=1}^{n}$, where each $g^{(i)} = (g_1^{(i)}, g_2^{(i)}, \ldots, g_t^{(i)})$. Here, the superscript $i$ indicates the index of the weight matrix to which the vector belongs, while the subscript $j$ (where $j$ ranges from 1 to $t$) denotes training iterations with each parameter. Following (Liu et al., 2019), we assume the adaptive learning rate of Adam $\psi(.) = \sqrt{\frac{1-\beta_2^t}{(1-\beta_2)\sum_{i=1}^{t}\beta_2^{t-i}g_i^2}}$, and $g_j^{(i)} \sim \mathcal{N}(0, \sigma^2)$ for all $i$ and $j$ in the early stage. Additionally, approximate the distribution of the exponential moving average as the distribution of the simple average, $p(\psi(.)) = p(\sqrt{\frac{1-\beta_2^t}{(1-\beta_2)\sum_{i=1}^{t}\beta_2^{t-i}g_i^2}}) \approx p(\sqrt{\frac{t}{\sum_{i=1}^{t}g_i^2}})$ (Nau, 2014), and then $\psi^2(.) \sim \text{Scale-inv-}\mathcal{X}^2(\rho, \frac{1}{\sigma^2})$.

**Theorem 2.** (Variance of Scaling Factors) In the early stages of training, if $\psi^2(\cdot) \sim \text{Scale-inv-}\mathcal{X}^2(\rho, \frac{1}{\sigma^2})$, and $g_j^{(i)} \sim \mathcal{N}(0, \sigma^2)^4$ for all $i, j$, then for all $\rho > 4$, the scaling factor $\phi^2 = \frac{\sum_{i=1}^{n}\psi_i^2(g_t^{(i)})^2}{\sum_{i=1}^{n}(g_t^{(i)})^2}$ satisfies $\text{Var}[\phi^2] \leq \text{Var}[\psi^2]$. If we approximate $\sqrt{\psi^2}$ and $\sqrt{\phi^2}$ to the first order, we have $\text{Var}[\phi] \leq \text{Var}[\psi]$.

*Proof.* We express $\phi^2$ as a weighted sum:

$$\phi^2 = \sum_{i=1}^{n} w_i \psi_i^2, \tag{26}$$

where the weights are defined as:

$$w_i = \frac{(g_t^{(i)})^2}{\sum_{j=1}^{n}(g_t^{(j)})^2}. \tag{27}$$

Each $w_i$ is a non-negative random variable satisfying $\sum_{i=1}^{n} w_i = 1$.

In the context of adaptive optimization algorithms like Adam, the squared gradients $\psi_i^2$ accumulate information from past iterations to adapt the learning rate for each parameter. With $\beta_2 = 0.999$, the moving average of the squared gradients places significant weight on historical data, making $\psi_i^2$ dependent mainly on past gradients, yielding:

$$\psi_i^2 \approx \psi_i^2(g_1^{(i)}, \ldots, g_{t-1}^{(i)}). \tag{28}$$

Since $\psi_i^2$ primarily depend on past gradients $g_1^{(i)}, \ldots, g_{t-1}^{(i)}$, and $w_i$ depend solely on the current gradients $g_t^{(i)}$, we can consider $\psi_i^2$ and $w_i$ to be independent random variables.

---

[4]The assumption of a mean-zero normal distribution is valid at the outset of training, as the weights are sampled from normal distributions with a mean of zero (Balduzzi et al., 2017)

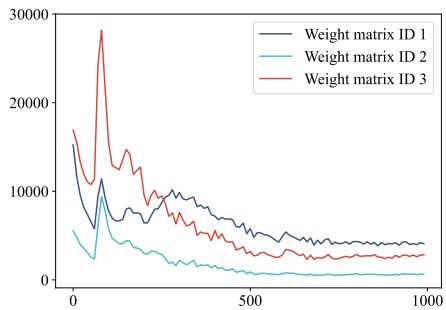 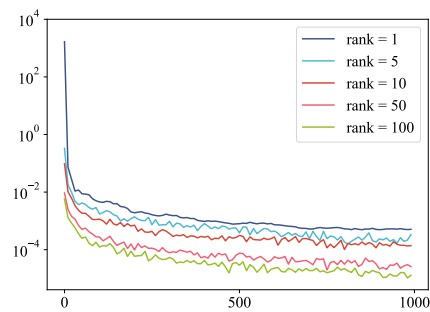

Figure 7: Scaling factor $\phi_t(R_t)$ during the early stage of training (1K iterations of total 10K iterations).

Figure 8: The simulation of variance of the scaling factor $\text{Var}[\phi]$ across different rank settings. The adaptive learning rate $\psi$ is equivalent to $\phi$ when the rank equals 1.

Consequently, we can express the variance of $\phi^2$ as:

$$\text{Var}[\phi^2] = \text{Var}\left[\sum_{i=1}^{n} w_i \psi_i^2\right]. \tag{29}$$

Using the law of total variance, we have:

$$\text{Var}[\phi^2] = \text{E}\left[\text{Var}\left[\sum_{i=1}^{n} w_i \psi_i^2 \mid w_1, \ldots, w_n\right]\right] + \text{Var}\left(\text{E}\left[\sum_{i=1}^{n} w_i \psi_i^2 \mid w_1, \ldots, w_n\right]\right). \tag{30}$$

Since $\psi_i^2$ are independent of the $w_i$, we find:

$$\text{E}\left[\sum_{i=1}^{n} w_i \psi_i^2 \mid w_1, \ldots, w_n\right] = \text{E}[\psi_i^2]\sum_{i=1}^{n} w_i = \text{E}[\psi_i^2], \tag{31}$$

$$\text{Var}\left[\sum_{i=1}^{n} w_i \psi_i^2 \mid w_1, \ldots, w_n\right] = \sum_{i=1}^{n} w_i^2 \text{Var}[\psi_i^2]. \tag{32}$$

Thus, the variance simplifies to:

$$\text{Var}[\phi^2] = \text{Var}[\psi^2]\,\text{E}\left[\sum_{i=1}^{n} w_i^2\right]. \tag{33}$$

The second term, $\text{Var}\left(\text{E}\left[\sum_{i=1}^{n} w_i \psi_i^2 \mid w_i\right]\right)$, is zero since $\text{E}[\phi^2 \mid w_i] = \text{E}[\psi_i^2]$ is constant.

Let $X_i = (g_t^{(i)})^2$, where each $X_i \sim \sigma^2 \chi_1^2$. Then, we can express the weights as:

$$w_i = \frac{X_i}{\sum_{j=1}^{n} X_j}. \tag{34}$$

Since $X_i/\sigma^2 \sim \chi_1^2$, and each $w_i$ is the ratio of $X_i$ to the sum of all $X_j$, the vector $(w_1, \ldots, w_n)$ follows a Dirichlet distribution with parameters $\alpha_i = \frac{\nu_i}{2} = \frac{1}{2}$, where $\nu_i = 1$ is the degrees of freedom of $\chi_1^2$.

For a Dirichlet distribution, the expected value of $w_i^2$ is given by:

$$\text{E}[w_i^2] = \frac{\alpha_i(\alpha_i + 1)}{\left(\sum_{k=1}^{n} \alpha_k\right)\left(\sum_{k=1}^{n} \alpha_k + 1\right)}. \tag{35}$$

Substituting $\alpha_i = \frac{1}{2}$ and $\sum_{k=1}^{n} \alpha_k = \frac{n}{2}$ yields:

$$\text{E}[w_i^2] = \frac{\frac{1}{2}\cdot\frac{3}{2}}{\frac{n}{2}\cdot\left(\frac{n}{2}+1\right)} = \frac{3}{4}\cdot\frac{4}{n(n+2)} = \frac{3}{n(n+2)}. \tag{36}$$

Thus, summing over all $i$ gives:

$$\mathrm{E}\left[\sum_{i=1}^{n} w_i^2\right] = n \cdot \mathrm{E}[w_i^2] = \frac{3}{n+2}. \tag{37}$$

Finally, substituting this result back into the variance expression:

$$\mathrm{Var}[\phi^2] = \mathrm{Var}[\psi^2] \cdot \frac{3}{n+2}. \tag{38}$$

Since $n \geq 1$, it follows that:

$$\frac{3}{n+2} \leq 1, \tag{39}$$

which implies:

$$\mathrm{Var}[\phi^2] \leq \mathrm{Var}[\psi^2]. \tag{40}$$

Given $\rho > 4$ and $\psi^2(\cdot) \sim \text{Scale-inv-}\mathcal{X}^2(\rho, \frac{1}{\sigma^2})$, the variance of $\psi^2(\cdot)$ exists (Liu et al., 2019).

Since $\psi_i^2$ and $w_i$ are independent and $\sum_{i=1}^{n} \mathrm{E}[w_i] = 1$:

$$\mathrm{E}[\phi^2] = \mathrm{E}\left[\sum_{i=1}^{n} w_i \psi_i^2\right] = \sum_{i=1}^{n} \mathrm{E}[w_i]\,\mathrm{E}[\psi_i^2] = \mathrm{E}[\psi_i^2] \sum_{i=1}^{n} \mathrm{E}[w_i] = \mathrm{E}[\psi_i^2], \tag{41}$$

Thus, we have shown that:

$$\mathrm{Var}[\phi^2] \leq \mathrm{Var}[\psi^2], \quad \text{and} \quad \mathrm{E}[\phi^2] = \mathrm{E}[\psi_i^2]. \tag{42}$$

Follow Liu et al. (2019), we approximate $\sqrt{\psi^2}$ and $\sqrt{\phi^2}$ to the first order (Wolter & Wolter, 2007)

$$\mathrm{Var}[\psi] \approx \frac{\mathrm{Var}[\psi^2]}{4\,\mathrm{E}[\psi^2]}, \quad \text{and} \quad \mathrm{Var}[\phi] \approx \frac{\mathrm{Var}[\phi^2]}{4\,\mathrm{E}[\phi^2]}. \tag{43}$$

which implies:

$$\mathrm{Var}[\phi] \leq \mathrm{Var}[\psi]. \tag{44}$$

$\square$

To further examine our theorem, we conduct simulations to calculate the variance of the scaling factor $\phi$ at ranks within the set $\{1, 5, 10, 50, 100\}$. The adaptive learning rate $\psi$ is equivalent to that of $\phi$ when the rank equals 1. As shown in Figure 8, the variance decreases as the rank increases, supporting our above theorem $\mathrm{Var}[\phi] \leq \mathrm{Var}[\psi]$. Furthermore, we observe a surprisingly large variance during the early stage, which corroborated our initial experiments. Consequently, we conclude that our method is efficient without requiring an additional warm-up.

## A.2 DETAILED PRE-TRAINING SETTING

This section provides an overview of the LLaMA architectures and the hyper-parameters employed during pre-training. To ensure a fair comparison, we adopt the same settings as Zhao et al. (2024a). Table 8 presents the hyper-parameters of the LLaMA architectures across various sizes. For all architectures, we utilize a maximum sequence length of 256 and a batch size of 131K tokens. Furthermore, we implement a learning rate warm-up during the initial 10% of training steps and employ cosine annealing for the learning rate schedule, which decreases to 10% of the initial learning rate.

For all methods except Fira and GaLore, we tune the optimal learning rate from the set $\{0.01, 0.005, 0.001, 0.0005, 0.0001\}$ across model sizes ranging from 60M to 1B, selecting their best validation perplexity to report. In contrast, both Fira and GaLore employ the same learning rate 0.01 and a subspace change frequency $T$ of 200 without tuning. Additionally, the scale factor $\alpha$ is considered a fractional learning rate (Zhao et al., 2024a). Furthermore, a relatively large learning rate may result in spikes of the training loss (Zhao et al., 2024a). To address this issue, for models with a size of less than 1B, we set $\alpha$ to 0.25, while for models exceeding 1B, we adjust $\alpha$ to 0.0625.

Table 8: Hyper-parameters of LLaMA architectures for pre-training.

| Params | Hidden | Intermediate | Heads | Layers | Steps | Data Amount (Tokens) |
|--------|--------|--------------|-------|--------|-------|----------------------|
| 60M    | 512    | 1376         | 8     | 8      | 10K   | 1.3 B                |
| 130M   | 768    | 2048         | 12    | 12     | 20K   | 2.6 B                |
| 350M   | 1024   | 2736         | 16    | 24     | 60K   | 7.8 B                |
| 1 B    | 2048   | 5461         | 24    | 32     | 100K  | 13.1 B               |
| 7 B    | 4096   | 11008        | 32    | 32     | 150K  | 19.7 B               |

Table 9: Hyper-parameter configurations of fine-tuning LLaMA-7B for Fira.

| Hyper-parameters | Setting |
|------------------|---------|
| Rank $r$         | 32      |
| $\alpha$         | 64      |
| Dropout          | 0.05    |
| Base optimizer   | AdamW   |
| LR               | 1e-4    |
| LR Scheduler     | Linear  |
| Batch size       | 16      |
| warm-up Steps    | 100     |
| Epochs           | 3       |
| Where            | Q,K,V,Up,Down |

### A.3 Detailed Fine-tuning Setting

We fine-tune the pre-trained LLaMA-7B model for commonsense reasoning tasks benchmark designed for LLM fine-tuning, which include eight sub-tasks (Hu et al., 2023). Table 9 shows the hyper-parameter configurations.

### A.4 Plug-and-play framework for Fira

---
**Algorithm 2** Plug-and-play framework for Fira, Pytorch-like.

---
1: **for** weight in model.parameters() **do**
2:     grad = weight.grad
3:     sub_grad, outer_grad = **project**(grad)                              ▷ Gradient projection.
4:     sub_adapt = **adapt**(sub_grad)                  ▷ **Adaptive optimizer**, e.g., Adam, RMSProp.
5:     outer_Fira = **Fira**(sub_grad, sub_adapt, outer_grad)          ▷ Apply Fira to outer_grad.
6:     weight_update = **project_back**(sub_grad) + outer_Fira           ▷ full-rank training
7:     weight.data += weight_update
8: **end for**

---

### A.5 Case Quantitative Analysis of Scaling Factor Similarities

In this section, we analyze the similarities among the three ranking (i.e., order) sequences $R_1$, $R_2$, and $R_3$ depicted in Figure 3, which highlight the discrepancies in scaling factors across different weight matrices. Each ranking comprises ten distinct items. We will employ Kendall's Tau correlation coefficient (Abdi, 2007) and Spearman's rank correlation coefficient (Sedgwick, 2014) to evaluate their concordance and divergence.

The three ranking sequences are defined as follows:

$$R_1 = (7, 6, 1, 2, 4, 8, 5, 10, 9, 3) \tag{45}$$

$$R_2 = (7, 8, 2, 1, 5, 4, 6, 10, 9, 3) \tag{46}$$

$$R_3 = (6, 8, 2, 1, 5, 4, 7, 10, 9, 3) \tag{47}$$

### A.5.1 KENDALL'S TAU CORRELATION COEFFICIENT

Kendall's Tau, a non-parametric statistic, measures the ordinal association between two rankings. It quantifies the degree to which the presence of one ranking implies a similar ranking in another. The formula for Kendall's Tau is given by:

$$\tau = \frac{C - D}{\frac{n(n-1)}{2}} \tag{48}$$

where:

- $C$ is the number of concordant pairs, which are pairs of observations where the ranks for both items agree,

- $D$ is the number of discordant pairs, where the ranks disagree,

- $n$ is the total number of observations.

Kendall's Tau ranges from -1 to +1, with 1 indicating perfect agreement between the rankings, 0 indicating no correlation, and -1 indicating perfect disagreement. This measure is particularly robust against outliers, making it useful for assessing the strength of relationships in ordinal data. A p-value, calculated to evaluate the statistical significance of the observed correlation, tests the null hypothesis that there is no association between the two variables. A low p-value (typically less than 0.05) suggests rejecting the null hypothesis, indicating that the observed correlation is statistically significant.

### A.5.2 SPEARMAN RANK CORRELATION COEFFICIENT

The Spearman rank correlation coefficient is another non-parametric measure that assesses the strength and direction of association between two ranked variables. It is calculated as follows:

$$\rho = 1 - \frac{6 \sum d_i^2}{n(n^2 - 1)} \tag{49}$$

where:

- $d_i$ represents the differences between the ranks of each observation,

- $n$ is the number of observations.

Spearman's coefficient also ranges from -1 to +1, with similar interpretations as Kendall's Tau. A coefficient of 1 indicates perfect positive correlation, -1 indicates perfect negative correlation, and 0 indicates no correlation. Spearman's method excels when analyzing datasets that fail to meet the normality assumptions requisite for parametric tests. To assess the significance of the Spearman correlation, a p-value is calculated alongside the coefficient. This p-value tests the null hypothesis of no correlation between the rankings of the variables. A small p-value (often less than 0.05) indicates a statistically significant correlation, providing strong evidence against the null hypothesis.

### A.5.3 RESULTS

The results of the Spearman and Kendall correlation coefficients are summarized in Table 10:

Table 10: Spearman and Kendall correlation coefficients with corresponding p-values.

| Sample | Spearman | | Kendall | |
|---|---|---|---|---|
| | Coefficient | P-value | Coefficient | P-value |
| $R_1R_2$ | 0.8545 | 0.0016 | 0.7333 | 0.0022 |
| $R_1R_3$ | 0.8303 | 0.0029 | 0.6889 | 0.0047 |
| $R_2R_3$ | 0.9879 | 9.31e-08 | 0.9556 | 5.51e-06 |

### A.5.4 ANALYSIS AND CONCLUSIONS

The correlation analysis indicates strong positive relationships among the samples, as summarized below:

- For $R_1R_2$: The Spearman coefficient is 0.8545 (p = 0.0016) and the Kendall coefficient is 0.7333 (p = 0.0022), both indicating strong and statistically significant correlations.

- For $R_1R_3$: The Spearman coefficient of 0.8303 (p = 0.0029) and the Kendall coefficient of 0.6889 (p = 0.0047) further confirm significant positive associations.

- For $R_2R_3$: The Spearman coefficient of 0.9879 (p = 9.31e-08) and the Kendall coefficient of 0.9556 (p = 5.51e-06) suggest an almost perfect correlation, accompanied by high statistical significance.

In general, a p-value less than 0.05 indicates strong evidence against the null hypothesis, suggesting that the observed correlations are statistically significant and unlikely to have occurred by chance.

Overall, all samples exhibit strong correlations. The consistency across both correlation methods underscores the reliability of these findings, suggesting robust relationships that warrant further exploration.

### A.6 LARGE-SCALE QUANTITATIVE ANALYSIS OF SCALING FACTOR SIMILARITIES

In the previous section, we conduct a case quantitative analysis of the similarity of scaling factors between low-rank and full-rank training across 10 weight matrices of the LLaMA 60M model. To further substantiate our findings, we expand the scope of our experiment to include all matrices of LLaMA models ranging from 60M to 1B. Additionally, we add value-based metrics of similarity (e.g., cosine similarity, mean squared error (MSE), and Pearson's correlation coefficient), beyond original order-based metrics. We use the same low-rank setup as in Table 2. Then, we train these models and assess the similarity of scaling factors averaged over 10,000 steps. Additionally, to evaluate the effectiveness of a column-level fine-grained strategy for scaling factors, we perform a column-level quantitative similarity analysis. Due to the computational challenges posed by the large number of columns, we randomly sample 100 columns for each matrix for analysis. Specifically, in the LLaMA 1B model, over 10,000 columns are sampled.

Both Spearman, Kendall, and Pearson correlation coefficients range from -1 to +1. A coefficient of 1 signifies a perfect positive correlation, and -1 signifies a perfect negative correlation. The p-value helps us determine whether the observed correlation is statistically significant or if it could have occurred by random chance. For instance, a p-value less than 0.05 means there is less than a 5% probability that the observed correlation happened by chance if there was actually no correlation. Generally, a p-value below 0.05 suggests that a significant correlation exists. The cosine similarity score also ranges from -1 to 1. Vectors with scores close to 1 are very similar. As for MSE, the smaller the value the higher similarity. As shown in Table 11 and 12, we can observe the significant similarity of scaling factors between low-rank and full-rank LLM training (all coefficient and cosine similarity close to 1, while p-value and MSE close to 0). Thus, it is likely that the observed behavior is an inherent feature of LLM training, manifesting across a broad range of scenarios. This insight provides a robust experimental basis for our proposed norm-based scaling in Fira and helps explain its effectiveness.

Table 11: Spearman, Kendall, and Pearson correlation coefficients (p-values) at both the Matrix and Column levels for pre-training LLaMA models ranging from 60M to 1B parameters, averaged over 10,000 steps.

| Size | Matrix Level | | | Column Level | | |
|------|--------------|--------------|----------------|--------------|--------------|--------------|
| | Spearman | Kendall | Pearson | Spearman | Kendall | Pearson |
| 60M | 0.9972 (2e-62) | 0.9662 (7e-26) | 0.9891 (1e-46) | 0.9372 (0.0) | 0.7942 (0.0) | 0.8723 (0.0) |
| 130M | 0.9925 (2e-76) | 0.9409 (9e-37) | 0.9813 (2e-60) | 0.8698 (0.0) | 0.6830 (0.0) | 0.7805 (0.0) |
| 350M | 0.9770 (3e-113) | 0.8848 (5e-65) | 0.9766 (1e-112) | 0.9091 (0.0) | 0.7400 (0.0) | 0.8272 (0.0) |
| 1B | 0.9469 (1e-83) | 0.8249 (1e-56) | 0.9457 (6e-83) | 0.8331 (0.0) | 0.6513 (0.0) | 0.8112 (0.0) |

Table 12: Cosine Similarity and MSE at both the Matrix and Column levels for pre-training LLaMA models ranging from 60M to 1B parameters, averaged over 10,000 steps.

| Size | Matrix Level | | Column Level | |
|------|-------------------|-------|-------------------|-------|
| | Cosine Similarity | MSE | Cosine Similarity | MSE |
| 60M | 0.9922 | 3e-04 | 0.9273 | 3e-05 |
| 130M | 0.9901 | 2e-04 | 0.9046 | 2e-05 |
| 350M | 0.9893 | 1e-04 | 0.9174 | 1e-05 |
| 1B | 0.9795 | 2e-04 | 0.9229 | 1e-05 |

### A.7 ADDITIONAL ANALYSIS OF SPIKES

Maintaining the direction of the raw gradient without correction might be unable to effectively deal with the steep loss landscapes of LLM training like Adam (Zhang et al., 2020). The steep loss landscapes are likely to cause abrupt increases in raw gradients. When the raw gradients increase abruptly, the gradients' norm after norm-based scaling may also increase abruptly, as illustrated in Figure 4. This arises from the fact that the norm-based scaling method only adjusts the average gradient norm of the gradient at the matrix level, failing to make fine-grained adjustments to each parameter, unlike the optimizer Adam. As a result, a significant parameter update may occur, undermining previous optimization efforts, i.e. training loss spikes (Goodfellow et al., 2016; Zhang et al., 2020).

There are a lot of methods also proposed to solve loss spiking and stabilize training, e.g., embedding normalization (Le Scao et al., 2022), gradient shrink (Zeng et al., 2023), tensor-wise scaling (Dettmers et al., 2022). However, it is crucial to clarify that our Fira does not conflict these stabilization methods. For instance, when training the LLaMA model with Fira, it inherently incorporates stabilization methods like RMSNorm. Our norm-growth limiter is mainly aimed at addressing the gradient stability capability that our norm-based scaling method lacks compared to Adam. As shown in Figure 9, when we directly use Adam to pre-train the llama model, there will be no loss spike. However, since Fira maintains the original direction of the raw gradient ($G_t - P_t R_t$), similar to SGD, it may lack the capability to navigate the sharp loss landscapes in LLM training, thus leading to an additional loss spike.

In addition, for more comprehensive comparisons of our norm-growth limiter, we design two additional gradient stabilization variants to solve the loss spike: Gradient Shrink ($|S_t| = |S_t| \cdot \alpha + |S_{t-1}| \cdot (1 - \alpha)$), and Tensor-Wise Scaling ($|S_t| = |S_t| \cdot \alpha$), where $S_t = \phi_t(R_t)(G_t - P_t R_t)$ is the corrected gradient by applying our norm-based scaling. As shown in Figure 10 and Table 13, Fira outperforms other gradient stabilization methods. For further analysis, Gradient Shrink fails to solve the loss spike, while Tensor-Wise Scaling solves the loss spike but led to sub-optimal results.

Table 13: Validation perplexity (↓) of Fira across different gradient stabilization methods.

| Method | Ours | Gradient Shrink | Tensor-Wise Scaling | Gradient Clipping | Without Limiter |
|---|---|---|---|---|---|
| Perplexity (↓) | **31.06** | 33.98 | 33.81 | 31.22 | 32.22 |

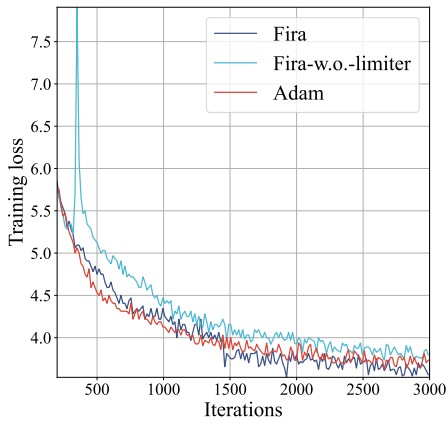

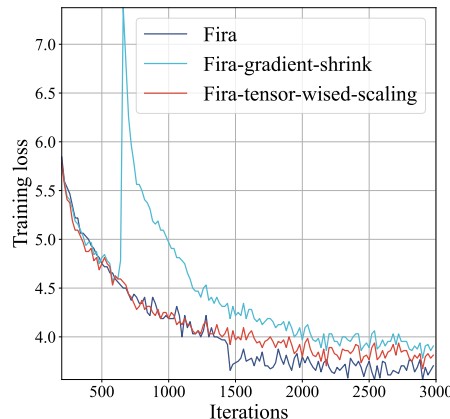

Figure 9: Training loss comparison of Adam, Fira, and Fira without limiter.

Figure 10: Training loss comparison of different gradient stabilization variant of Fira.

### A.8 MEMORY ESTIMATES.

Due to the difficulties associated with directly measuring GPU memory usage for a specific component, we estimate the memory requirements for weight parameters and optimizer states across various methods and model sizes (Zhao et al., 2024a). This estimate is derived from the number of parameters and optimizer states in BF16 format. In particular, for the memory of parameters, we multiply the total number of parameters by 2; for the memory of optimizer states, we first calculate the total number of optimizer states according to Table 1 and then multiply this total number by 2.

### A.9 ADDITIONAL EXPERIMENTS ON OVERHEAD.

We conduct additional comparisons regarding real memory usage and throughput of different memory-efficient training methods for both pre-training and fine-tuning. As illustrated in Tables 14 and 15, Fira achieves superior memory efficiency compared to full-rank training without significantly reducing throughput. Although Fira's throughput is slightly lower than that of other memory-efficient methods, it delivers exceptional performance. During pre-training, methods like LoRA necessitate maintaining higher-rank adapters compared to full-rank training. In practice, maintaining these higher-rank adapters outweighs the benefits of fewer trainable parameters, thus leading to more memory and less throughput. Furthermore, since full fine-tuning of LLaMA 7B's memory requirements exceeds the A100's 80GB capacity, we utilize DeepSeed's Zero2 technology to mitigate its memory usage.

Table 14: Real memory usage and normalized throughput when pre-training LLaMA 1B on the C4 dataset.

| Method | Fira | Galore | Flora | LoRA | ReLoRA | Full-rank |
|---|---|---|---|---|---|---|
| Memory (GB) | 54.6 | 54.6 | 54.5 | 59.0 | 59.0 | 58.5 |
| Normalized Throughput (%) | 94.2 | 95.9 | 95.9 | 67.4 | 67.4 | 100 |

Table 15: Real memory usage and normalized throughput when fine-tuning LLaMA 7B on commonsense reasoning datasets.

| Method | Fira | Galore | Flora | LoRA | ReLoRA | Full-rank |
|---|---|---|---|---|---|---|
| Memory (GB) | 23.4 | 23.4 | 23.3 | 23.7 | 23.7 | >80 |
| Normalized Throughput (%) | 156.1 | 201.1 | 210.3 | 232.8 | 232.8 | 100 |

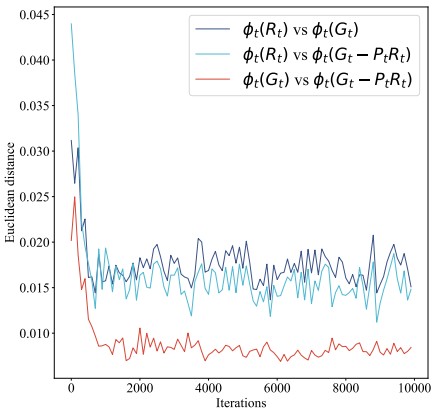

Figure 11: Euclidean distance trends over training iterations ($r/d_{model} = 128/256$).

Figure 12: Euclidean distance trends over training iterations ($r/d_{model} = 16/256$).

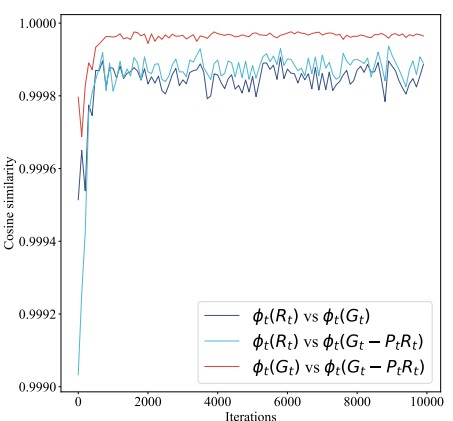
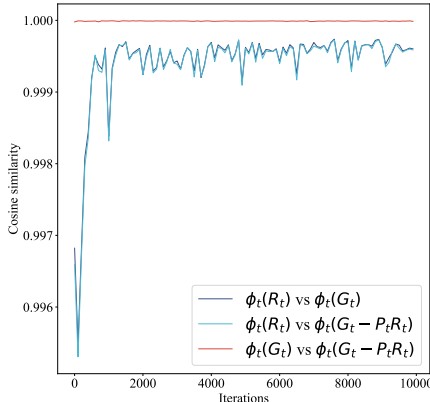

Figure 13: Cosine similarity trends over training iterations ($r/d_{model} = 128/256$).

Figure 14: Cosine similarity trends over training iterations ($r/d_{model} = 16/256$).

## A.10 ADDITIONAL EXPERIMENTS ON THE SIMILARITY TRENDS.

We conduct additional experiments on similarity trends of scaling factors from the same full-rank Adam dynamic using two rank settings: $16/256$ and $128/256$.

As shown in Figure 11, 12, 13, and 14, the similarity exhibits fluctuations during the initial training phase but achieves a relatively steady pattern with high similarity in the later iterations. Under lower rank setting $r/d_{model} = 16/256$, there is negligible reduction in similarity. Besides, $\phi_t(G_t)$ and $\phi_t(G_t - P_t R_t)$ demonstrate significantly higher similarity owing to their closely aligned dimensions.

This observation further validates that $\phi_t(R_t) \approx \phi_t(G_t) \approx \phi_t(G_t - P_t R_t)$.

### A.11 ADDITIONAL COMPARISONS OF PERPLEXITY TRENDS.

In this section, we compare the perplexity trends of Fira, Fira-only-scaling, SGD, and Adam. As illustrated in 15 (a), the performance of vanilla SGD is significantly inferior, highlighting its inadequacy for directly training LLMs. As depicted in 15 (b) and (c), while Adam demonstrates faster convergence during the initial stages, both Fira and Fira-only-scaling achieve superior performance in the later stages. This maybe because Fira applies an adaptive strategy only at the matrix-level while maintaining the original gradient direction within a weight matrix. In this way, Fira may introduce a higher degree of randomness in training and a better ability to escape the local optima.

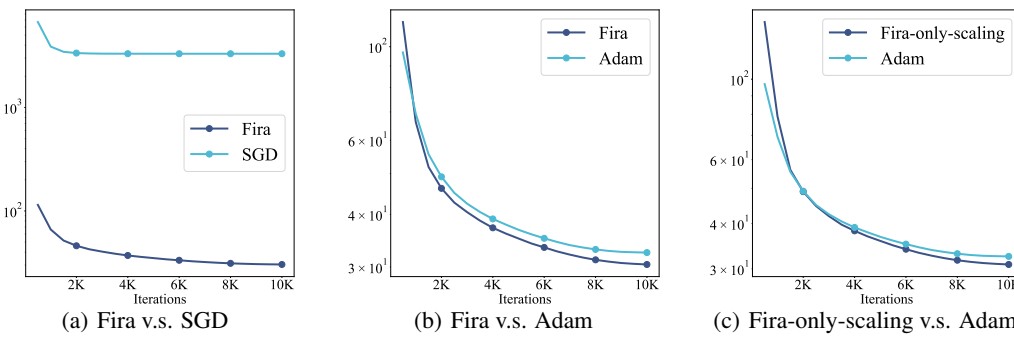

(a) Fira v.s. SGD      (b) Fira v.s. Adam      (c) Fira-only-scaling v.s. Adam

Figure 15: Comparisons of perplexity (↓) trends for pre-training LLaMA 60M on C4 dataset.

