# OpenReview forum: "Fira: Can We Achieve Full-rank Training of LLMs Under Low-rank Constraint?"
_ICLR.cc/2025/Conference — Submitted to ICLR 2025_

### Official Review · Reviewer_qp5x · 2024-10-27

**Soundness:** 2
**Presentation:** 1
**Contribution:** 3
**Rating:** 6
**Confidence:** 2

**Summary:**

This work considers training LLMs with full-rank gradients and weights within the low-rank constraints (i.e. imposing low-rank structures on gradient or weights). The main goal of this work is to improve Galore (a low-rank gradient projection optimizer [1]) by incorporating the orthogonal complement information of the Galore projected gradient, without maintaining its optimizer states (1st and 2nd order momentums).

Specifically, the authors observe that (Figure 3), when training LlaMA-60M on C4 dataset with Galore, the scaling factor (the ratio between the norm of the corrected gradient an original gradient) of each matrix exihibits similar rank orders across different choices of GaLore-rank $r$. This motivates the authors to extend GaLore to Fira, a full-rank weight & gradient optimizer under low-rank constraint that consists of two components: 1) a norm-based scaling method, which replace the full-rank scaling factor with the low-rank counterpart; 2) a norm-growth limiter to smooth the training gradient by restricting the gradient norm's increase. The authors showed that proposed Fira outperforms the full-rank baselines on both pretraining and finetuning datasets across various settings.

[1]. *Jiawei Zhao, Zhenyu Zhang, Beidi Chen, Zhangyang Wang, Anima Anandkumar, and Yuandong Tian. Galore: Memory-efficient llm training by gradient low-rank projection. ICML 2024.*

**Strengths:**

1. Compared to the Galore, this paper proposed to utilize the scaled gradient information (Algorithm 1, line 15) resides in the orthogonal complement of the Galore gradient, which achieves improved training performance. The utilization of the orthogonal complement gradient information is new.
2. The authors proposed a new 'gradient clipping' scheme, i.e. the norm-growth limiter in Eq. (12) to alleviate the spikes arouse in during the Fira training process.
3. The authors show evident outperformance of Fira compared to the standard Galore within a similar parameter amount (Table 1 and Table 2).

**Weaknesses:**

My mainly concerns lies in the motivation of using the norm-based scaling Eq. (10).

The authors observed that, the average scaling factors (averaged every 20 steps) of the weight matrices exihibit similar orders across different Galore-rank-$r$ training settings (Figure 3 and Table 7). The authors also mentioned that 'the absolute magnitude of the scaling factors can be regarded as the learning rate to which Adam is not sensitive' (Page 5, lines 266-267).

However, from my perspective, several logical gaps still remain between this observation and the 'utilization of norm-based scaling in Eq. (10) at every training step':

1. To approximate the effect of $(G_t-P_tR_t)$, according to the anaysis and discussions in Page 5 line 238-240, we want to scale it to approximate $\psi_t(G_t-P_tR_t)$ or $\psi_t(G_t-P_t\psi_t(R_t))$, that is, the full-rank gradient dynamic with optimizer states of the orthogonal complement or the full-gradient. But the similarity of the scaling factor orders of $\phi_t(R_t)$ with different Galore-ranks (Figure 3 and Table 7) does not sufficiently implies $\phi_t(R_t)\approx \phi_t(G_t-P_tR_t)$ or $\phi_t(R_t)\approx \phi_t(G_t-P_t\psi_t(R_t))$.
2. The scaling factor orders shown in Figure 3 and Table 7 only consider the top-10 weights matrices. Thus, this evidence does not sufficiently imply this 'stable order phenomenon' holds for all optimizable weight matrices.
3. Even if the order of scaling factors of all matrices are the same, this can not sufficiently imply the magnitude of the scaling factors can be absorbed into the learning rate (as mentioned in Page 5 line 267). This is because different matrices have different scaling factors, and the equivalent learning rates of all matrices are different.

According to Table 4, it seems that the performance gain mainly comes from this norm-based scaling. Therefore, it is necessary to identify the motivation and mechanism of how this works and what is the essential effect of this scaling effect.

**Questions:**

See **Weakness**.

---

> ### Author Response · Authors · 2024-11-23
> **Response to Reviewer qp5x (1/3)**
>
> Thanks for the reviewer's detailed feedback and comments. We have added source code in **Supplementary Material** for reproducibility and will provide the link to our GitHub repository upon acceptance. For the convenience of answering, we have advanced the responses for weaknesses 2 and 3. Below are our responses to the specific concerns you raised.
>
> >W2: The scaling factor orders shown in Figure 3 and Table 7 only consider the top-10 weights matrices. Thus, this evidence does not sufficiently imply this 'stable order phenomenon' holds for all optimizable weight matrices.
>
> >W3: Even if the order of scaling factors of all matrices are the same, this can not sufficiently imply the magnitude of the scaling factors can be absorbed into the learning rate (as mentioned in Page 5 line 267). This is because different matrices have different scaling factors, and the equivalent learning rates of all matrices are different.
>
> A2 & A3: Thanks for your valuable suggestion. To further verify the validity of our approximation, we have conducted **more comprehensive experiments** on the similarity of scaling factors between low-rank and full-rank LLM training in Appendix A.6:
> + We **expanded the scope of our experiments** to include **all optimizable weight matrices** of LLaMA models ranging from 60M to 1B and assessed the similarity of scaling factors averaged over 10,000 steps.
> + We added additional **value-based metrics** of similarity **beyond order-based metrics** (e.g., cosine similarity, mean squared error (MSE), and Pearson's correlation coefficient).
> + We additionally performed a **column-level** quantitative similarity analysis, to evaluate the effectiveness of our column-level fine-grained strategy for scaling factors.
>
> The results are as follows:
> Table r1: Spearman, Kendall, and Pearson correlation coefficient (p-value) at both matrix and column levels.
> | Size   | Matrix Level    |  |  | Column Level    |    | |
> |--------|---------------------------|---------------------------|---------------------------|---------------------------|---------------------------|--------------------------|
> |        | Spearman      | Kendall |Pearson   | Spearman     | Kendall     | Pearson |
> | 60M    | 0.9972 (2e-62)            | 0.9662 (7e-26)            | 0.9891 (1e-46)| 0.9372 (0.0)              | 0.7942 (0.0)             |0.8723 (0.0) |
> | 130M   | 0.9925 (2e-76)            | 0.9409 (9e-37)            |0.9813 (2e-60) |0.8698 (0.0)              | 0.6830 (0.0)             | 0.7805 (0.0)|
> | 350M   | 0.9770 (3e-113)           | 0.8848 (5e-65)           | 0.9766 (1e-112)|0.9091 (0.0)              | 0.7400 (0.0)             |0.8272 (0.0) |
> | 1B     | 0.9469 (1e-83)            | 0.8249 (1e-56)           |0.9457 (6e-83) |0.8331 (0.0)              | 0.6513 (0.0)             |0.8112 (0.0) |
>
> Table r2:  Cosine Similarity and MSE at both matrix and column levels.
> | Size   | Matrix Level    |  | Column Level    | |
> |--------|---------------------------|---------------------------|---------------------------|---------------------------|
> |        | Cosine Similarity    | MSE     | Cosine Similarity   | MSE |
> | 60M    | 0.9922 | 3e-04 | 0.9273 | 3e-05 |
> | 130M   | 0.9901 | 2e-04 | 0.9046 | 2e-05 |
> | 350M   | 0.9893 | 1e-04 | 0.9174 | 1e-05 |
> | 1B     | 0.9795 | 2e-04 | 0.9229 | 1e-05 |
>
> Thus, we can observe significant similarity in scaling factor between low-rank and full-rank LLM training at **both order and value levels** across **a broad range of scenarios** (all coefficient and cosine similarity close to 1, while p-value and MSE close to 0). This insight provides **a stable experimental basis** for our proposed norm-based scaling in Fira and helps explain the validity of this approximation.
>
> Moreover, based on this significant similarity of the relative value of scaling factors, we can **absorb its absolute magnitude into the overall learning rate**. We believe that the scaling factor is able to capture the inherent variability between weight matrices. Therefore, our method is actually an adaptive strategy at the matrix or column level according to the similarity of their scaling factors. Additionally, within a weight matrix, the low-rank gradient $R_t$ can essentially be regarded as a sampling of the full-rank gradient $G_t$. Thus we can approximately assume that $\phi_t(R_t)\approx \phi_t(G_t)$.

---

> ### Author Response · Authors · 2024-11-23
> **Response to Reviewer qp5x (2/3)**
>
> >W1: To approximate the effect of $(G_t-P_tR_t)$,  according to the anaysis and discussions in Page 5 line 238-240, we want to scale it to approximate $\psi_t(G_t-P_tR_t)$ or $\psi_t(G_t-P_t\psi_t(R_t))$, that is, the full-rank gradient dynamic with optimizer states of the orthogonal complement or the full-gradient. But the similarity of the scaling factor orders of $\phi_t(R_t)$ with different Galore-ranks (Figure 3 and Table 7) does not sufficiently implies $\phi_t(R_t) \approx \phi_t(G_t-P_tR_t)$ or $\phi_t(R_t) \approx \phi_t(G_t-P_t\psi_t(R_t))$.
>
>
> A1: We appreciate the insightful observation regarding the motivation of norm-based scaling. In the previous section, we have provided **sufficient experiments** to demonstrate the similarity of the scaling factor in **both order and value levels**. Based on this insight, we use low-rank scaling factors $\phi_t(R_t)$ to approximate full-rank scaling factors $\phi_t(G_t)$. For further analysis, we will:
> + Give **the error upper bound** on this approximation **theoretically** and **verify** our analysis **experimentally**.
> + Deduce that $\phi_t(R_t) \approx \phi_t(G_t-P_tR_t)$ given $\phi_t(R_t)\approx \phi_t(G_t)$.
>
> **Theoretical error upper bound on the approximation of norm-based scaling.**
>
> To quantify the error of our approximation, we have added **its theoretical error upper bound**, and **verified** our analysis **experimentally** in in Appendix A.1.
>
> $$ \left| \phi^2_t(G_t)-\phi^2_t(R_t)  \right| \leq (c_{\max}^2 - c_{\min}^2) \cdot (1-\frac{\sum_{i=1}^{r} g_i^2}{\sum_{i=1}^{n} g_i^2}),$$
>
> where $\phi_t(G_t)$, $\phi_t(R_t)$ are the scaling factors of full-rank and low-rank gradients respectively. We assume that scaling factors $\phi_t$ are bounded between constants $c_{\max}$, $c_{\min}$, and $g_i$ is the component of the gradients $g$. To simplify the proof, we consider that $r$ components of low-rank gradients are directly sampled from $n$ components of full-rank gradients. From this theory, we can find that the error upper bound on the approximation of scaling factors is mainly determined by two aspects:
>
> + Variability of Scaling Factor $(c_{\max}^2 - c_{\min}^2)$: This term represents the maximum variation in the scaling factors of different gradient components. For further validation, we designed **Fira-only-scaling**, a variant of Fira. It directly applies the low-rank scaling factors to the full-rank gradients by changing the Eq. (10) from $  W_{t+1}=W_{t} - \eta P_t\psi_t(R_t) - \eta \phi_t(R_t)  (G_t-P_tR_t)$ to $ W_{t+1}=W_{t} - \eta \phi_t(R_t)G_t$. In this way, we are able to exclude the influence of the original Adam term $P_t\psi_t(R_t)$ and better analyze the effectiveness of our approximation. As shown in Table r3, Fira (column-level) gains better performance than Fira (matrix-level) for its more fine-grained consideration of the scaling factor, which also means a smaller maximum variation $(c_{\max}^2 - c_{\min}^2)$.
>
> + Effectiveness of Gradient Sampling $(1-\frac{\sum_{i=1}^{r} g_i^2}{\sum_{i=1}^{n} g_i^2})$: This term represents the proportion of the gradients norm contributed by the sampled low-rank \( r \) components from full-rank \( n \) components. As shown in Table r4, we conducted ablation experiments **Fira-only-scaling-w.o.-svd**, i.e., Fira-only-scaling without SVD in low-rank gradient sampling.  As we can see, SVD is capable of sampling more prominent low-rank gradients, which leads to a reduction in the upper bound of error and enhanced performance. Similarly, as shown in Table r5, employing a higher rank enables the sampling of a greater proportion of the gradients norm, resulting in reduced error upper bound and improved performance.
>
> Furthermore, the slight reduction of our method with decreasing rank (e.g., Rank = 16) underscores the validity of our approximation.
>
> Table r3: Validation perplexity ($\downarrow$)  of Fira-only-scaling at Column-level and Matrix-level.
> | Column-level | Matrix-level |
> |------|------|
> | **31.68** | 32.05 |
>
> Table r4: Validation perplexity ($\downarrow$)  of Fira-only-scaling with and without SVD.
> | Fira-only-scaling | Fira-only-scaling-w.o.-svd |
> |------|------|
> |**31.68** | 32.22 |
>
> Table r5:  Validation perplexity ($\downarrow$) of Fira-only-scaling across different ranks.
> | Rank |   4  |  16  |  64  |  128 |
> |------|------|------|------|------|
> | Fira-only-scaling | 35.91 | 32.90 | 31.93 | **31.68** |

---

> ### Author Response · Authors · 2024-11-23
> **Response to Reviewer qp5x (3/3)**
>
> **Continuation of A1.**
>
> **Derivation of $\phi_t(R_t) \approx \phi_t(G_t-P_tR_t)$ given $\phi_t(R_t)\approx \phi_t(G_t)$.**
>
> Given that $G_t=(G_t-PR_t) + PR_t$, it follows that:
>
> $$||G_t||^2 = ||G_t-PR_t||^2 + ||PR_t||^2.$$
>
> Similarly, the gradients after the correction of adaptive optimizers can be decomposed as:
>
> $$||G_t||^2 \cdot \phi_t^2(G_t) = ||G_t-PR_t||^2 \cdot \phi_t^2(G_t-PR_t) +||PR_t||^2 \cdot \phi_t^2(PR_t). $$
>
> As we know, $R_t$ represents the original subspace gradient, and $PR_t$ is the same gradient which is only transformed into the full space via inverse projection. Theoretically, this projection does not change the gradient and does not affect the subspace optimization of the adaptive optimizer. Thus, $\phi(PR_t) \approx \phi_t(R_t)$.
>
> In our practice, we calculated the scaling factor using two approaches: before re-projection ($\phi(R_t)=\frac{||\psi(R_t)||}{||R_t||}$) and after re-projection ($\phi(PR_t)=\frac{||P\psi(R_t)||}{||PR_t||}$). The outcomes of experiments were very close.
>
> Then
> $$\phi^2_t(G_t-P_tR_t) = \frac{||G_t||^2 \cdot \phi_t^2(G_t)-||PR_t||^2 \cdot \phi_t^2(PR_t)}{||G_t-PR_t||^2}.$$
> Since $\phi_t(R_t)\approx \phi_t(G_t)$ and $\phi(PR_t) \approx \phi_t(R_t)$, we derive that:
> $$\phi^2_t(G_t-P_tR_t) \approx \phi^2_t(R_t) \cdot \frac{||G_t||^2 -||PR_t||^2 }{||G_t-PR_t||^2}.$$
> Since $||G_t||^2 = ||G_t-PR_t||^2 + ||PR_t||^2$, $\phi_t(G_t-P_tR_t)>0$, and $\phi_t(R_t)>0$, we have:
> $$\phi_t(G_t-P_tR_t) \approx \phi_t(R_t).$$
>
> Lastly, we express our sincere gratitude for your insightful comments. Regarding the motivation concerns raised, we will carefully incorporate all suggested improvements in our next revision. We hope that you will consider increasing your score if your concerns are adequately addressed. If you have any question, please feel free to point out and we will try to address it quickly.

---

> > ### Comment · Reviewer_qp5x · 2024-11-24
> >
> > Dear authors,
> >
> > I appreciate your time and effort in providing additional experiments showing that "$\phi_t(G_t)\approx \phi_t(R_t)$".
> >
> > However, I still have some confusion in the derivation of "$\phi_t(G_t)\approx \phi_t(R_t) \overset{?}{\implies}\phi_t(G_t)\approx \phi_t(G_t - P_tR_t)$".
> >
> > To my understanding, the key challenging in proving this "statement" is that, we need to prove the equality holds uniformly for all $t$.
> >
> > But in fact, the momentums of the sequences $R_t$ and $G_t - P_tR_t$ is not orthogonal, since $P_t$ keeps varying, and the subspace induced by $P_t$ and $P_s$ might not be the same, for $t \neq s$.
> >
> > Thus, the momentum terms $\\text{Momentum}(P_tR_t)$ does not resides in $\text{span}(P_t)$, and $\\text{Momentum}(G_t - P_t R_t)$ does not resides in $\text{span}(I - P_rP_t^\top)$. Therefore, the two terms might not be orthogonal.
> >
> > In general, this implies that
> > $\\|G_t\\|^2 \phi_t(G_t) = \\|\text{Momentum}(G_t)\\|^2 = \\| \\text{Momentum}(P_tR_t) + \text{Momentum}(G_t - P_t R_t)  \\|^2 \neq \\| \text{Momentum}(P_tR_t)\\|^2 + \\|\text{Momentum}(G_t - P_t R_t)  \\|^2$
> >
> > As I remain skeptical in the statement that: $\\|G_t\\|^2 \phi_t(G_t)=\\|R_t\\|^2 \phi_t(P_tR_t) + \\|G_t-P_tR_t\\|^2 \phi_t(G_t-P_tR_t)$, I have to keep my rating.

---

> > > ### Author Response · Authors · 2024-11-24
> > > **Thanks for your valuable feedback**
> > >
> > > Thanks for your valuable feedback. As you said, the key to prove the above "statement" is that the projection matrix **$P_t$ remains unchanged**.
> > > It is important to clarify that the projection matrix $P_t$ in our method remains unchanged for the majority of iterations. Following Galore [1], $P_t$ is updated only once every $T$ iterations ($T$ is usually above 200) and **is treated as constant in Galore** (see Theorem 3.8 in Galore). Consistent with Galore, we also consider it to be a fixed constant.
> > >
> > > Looking forward to further discussion with you.
> > >
> > > [1]. GaLore: Memory-Efficient LLM Training by Gradient Low-Rank Projection, In ICML 2024

---

> > > > ### Comment · Reviewer_qp5x · 2024-11-24
> > > >
> > > > Dear authors,
> > > >
> > > > Thanks for your further justification on the fact that "$P_t$ remains unchanged for an interval of length $200$".
> > > >
> > > > From my understanding, in theory, the conclusion derived from an unchanged $P_t$ only applies to the first $T$ training steps, and does not apply to the successive training steps (e.g. the $(10\times T)$-th step). This is because the early momentums (at least $T$ epochs earlier) are attached in the optimizer state, which can lead to violation of the "$P_t$ unchanged" assumption.
> > > >
> > > > Furthermore, in practice, $200$ is considered a small number compared to the length of the whole training process. As "updating the optimization subspace" is very crucial for GaLore-like algorithms to achieve good performance, a $P_t$-varying analysis becomes necessary in this case.
> > > >
> > > > In Theorem 3.8 of [1], their statement is confined to "GaLore with fixed $P_t$ tends to converges". Therefore, they can assume $P_t$ unchanged during the derivation. However, in this work, the variability of $P_t$, $R_t$, and $G_t - P_t R_t$ is the core of the rationale of designing "scalers or correctors". Thus, a $P_t$-varying analysis seems to be necessary for this work.

---

> ### Comment · Reviewer_qp5x · 2024-11-24
>
> Dear authors,
>
> While I appreciate the additional experiments in validating $\phi_t(R_t)\approx \phi_t(G_t)$, I still have some confusion regarding the experiment design and results.
>
> To my understanding, validating the equality $\phi_t(R_t)\approx \phi_t(G_t)$ requires us to calculate the difference or similarity between the vectors $v_1 = (\phi_t(R_t^1),...,\phi_t(R_t^L))$ and $v_2 = (\phi_t(G_t^1),...,\phi_t(G_t^L))$, where $R_t^l$ and $G_t^l$ are the low-rank- and full-rank-gradient of the $l$-th weight matrix. Notably, the $R_t$ and $G_t$ needs to be computed from the same training dynamic (parameter training trajectory), rather than computing $R_t$ from a GaLore-Adam dynamic and computing $G_t$ from a full-rank Adam dynamic. Only in this way, we can ensure $R_t$ is indeed the low-rank projected version of $G_t$. Otherwise, we are comparing the statistics from two different training dynamic, failing to obtain supportive evidence towards the motivation of designing Fira.
>
> Secondly, I am confused about why the authors assess the difference between the rank of $v_1$ and $v_2$ (via rank-test), instead of assessing the element-wise distance between them (i.e. computing the maximal deviation $\\|v_1- v_2\\|$). From my understanding, the conclusion $\phi_t(R_t)\approx \phi_t(G_t)$ only if it is observed that $\\|v_1- v_2\\| $ is sufficiently small. It seems that there is a logic gap between the "similarity of the rank of scalers" and "the similarity of the value of scalers".

---

> > ### Author Response · Authors · 2024-11-25
> > **Thanks for your valuable feedback**
> >
> > Thanks for your further feedback and suggestions. To convince you that $\phi_t(R_t)\approx \phi_t(G_t)\approx \phi_t(G_t-P_tR_t)$ does always hold, we conducted experiments as you suggested.
> > We calculated $(\phi_t(R_t^1),...,\phi_t(R_t^L))$, $(\phi_t(G_t^1),...,\phi_t(G_t^L))$, $(\phi_t((G_t-P_tR_t)^1),...,\phi_t((G_t-P_tR_t)^L))$ of **all weight matrices** from **the same full-rank Adam dynamic** with switching frequency $T=200$. Based on it, we calculated the Euclidean distance $\\|v_1-v_2\\|$ and the cosine similarity between each pair of terms above. The results averaged over 10,000 training steps are as follows:
> >
> > Table r6: $\\|v_1-v_2\\|$ and cosine similarity of different comparisons.
> > | Comparison | $\|\|v_1- v_2\|\|$, cosine similarity|
> > |--------|--------|
> > | $(\phi_t(R_t^1),...,\phi_t(R_t^L))$ vs   $(\phi_t(G_t^1),...,\phi_t(G_t^L))$|  0.014, 0.9999 |
> > | $(\phi_t(R_t^1),...,\phi_t(R_t^L))$ vs $(\phi_t((G_t-P_tR_t)^1),...,\phi_t((G_t-P_tR_t)^L))$  |0.011, 0.9999 |
> > | $(\phi_t(G_t^1),...,\phi_t(G_t^L))$ vs $(\phi_t((G_t-P_tR_t)^1),...,\phi_t((G_t-P_tR_t)^L))$  |0.006, 1.0000|
> >
> > We can find that this similarity is more significant when calculating from the same training dynamic and $T=200$ ($\\|v_1- v_2\\|$ is sufficiently small and cosine similarity is close to 1). This fully validates that $\phi_t(R_t)\approx \phi_t(G_t)\approx \phi_t(G_t-P_tR_t)$.
> >
> > Hope the above response can address your concerns. Looking forward to further discussion with you.

---

> ### Comment · Reviewer_qp5x · 2024-11-25
>
> Dear authors,
>
> Thanks for your additional experiment results. Admittedly, the reported average value $\\|v_1-v_2\\|$ is surprisingly (for me, personally) small. May I know what is the rank in this experiment?
>
> Furthermore, instead of reporting the averaged deviation, I believe visualizing how $\\|v_t - v_2\\|$ changes across the training step can help exploring to what extent does the equality $\phi_t(R_t)\approx \phi_t(G_t) \approx \phi_t(G_t - P_tR_t)$ hold.

---

> ### Author Response · Authors · 2024-11-25
> **Thanks for your valuable feedback**
>
> Thanks for your further feedback and suggestions. The rank setting of Table r6 is $128/256$. To further validate the similarity of scaling factors, we conducted additional experiments using two rank settings: $16/256$ and $128/256$. **The visualization of similarity trends over training iterations** is provided in Section A.10 on page 24 (Figure 11-14) of the updated manuscript's Appendix. We also provide the averaged results of different rank settings as follows:
>
> Table r7: $\\|v_1-v_2\\|$ and cosine similarity of different comparisons (the first 200 iterations were excluded due to fluctuations).
> | Comparison | $r/d_{model}=16/256$ | $r/d_{model}=128/256$ |
> |--------|--------|--------|
> | $(\phi_t(R_t^1),...,\phi_t(R_t^L))$ vs   $(\phi_t(G_t^1),...,\phi_t(G_t^L))$|  0.019, 0.9998 |0.014, 0.9999
> | $(\phi_t(R_t^1),...,\phi_t(R_t^L))$ vs $(\phi_t((G_t-P_tR_t)^1),...,\phi_t((G_t-P_tR_t)^L))$  |0.019, 0.9998 | 0.011, 0.9999
> | $(\phi_t(G_t^1),...,\phi_t(G_t^L))$ vs $(\phi_t((G_t-P_tR_t)^1),...,\phi_t((G_t-P_tR_t)^L))$  |0.001, 1.0000| 0.006, 1.0000 |
>
>
> As shown in the figures and table, the similarity exhibits fluctuations during the initial training phase but achieves a relatively steady pattern with high similarity in the later iterations. Under lower rank setting $r/d_{model}=16/256$, there is negligible reduction in similarity. Besides, $\phi_t(G_t)$ and $\phi_t(G_t-P_tR_t)$ demonstrate significantly higher similarity owing to their closely aligned dimensions.
>
> This observation further validates that $\phi_t(R_t)\approx \phi_t(G_t)\approx \phi_t(G_t-P_tR_t)$.

---

> ### Comment · Reviewer_qp5x · 2024-11-25
>
> Dear authors,
>
> Thanks for your added experiments. I will raise my score to 6.
>
> In particular, I found the rebuttal response to other reviewers are not very convincing. I raise my score mainly because the experiment results seem to be surprisingly strong. At this point, I cannot verify the correctness of the results. From my perspective, the writing of this paper has a large room for improvement.

---

> > ### Author Response · Authors · 2024-11-25
> >
> > Thank you for raising your score! We will further incorporate your suggested improvements in the final version.

---

### Official Review · Reviewer_Vz4X · 2024-11-03

**Soundness:** 3
**Presentation:** 2
**Contribution:** 3
**Rating:** 6
**Confidence:** 4

**Summary:**

This paper proposes a new method to achieve efficient training of LLMs by reducing the memory usage for optimizer states. Specifically, it designs a norm-based scaling method that approximates full-rank training by using the scaling impact of low-rank optimizers as substitutes for that of original full-rank optimizers. It also proposes a norm-growth limiter to address the issue of loss spikes.

**Strengths:**

The proposed method demonstrates strong empirical performance. While the training efficiency is close to the SOTA method, GaLora, the proposed method significantly outperforms GaLora even using 8X smaller ranking. Moreover, it can even outperform full-rank training.

**Weaknesses:**

1. The authors support the use of low-rank scaling factors to approximate full-rank scaling factors by demonstrating that the order of the average scaling factors of the weight matrix is highly similar. However, similarity in order is not equivalent to similarity in value. Therefore, the validity of this approximation method is questionable.

2. Since the proposed method aims to approximate full-rank training, we would assume that the performance of full-rank training is the upper bound. However, the proposed method even outperforms full-rank training. It is unclear why the proposed method outperforms.

3. If the scaling factor for the full rank gradient can be approximated by a low rank scaling factor, why not just apply the low rank scaling factor to the full rank gradient? This strategy has the same memory usage in the optimizer state as the proposed method. Could the authors conduct experiments to validate this strategy? This would be a good support for the effectiveness of the scaling factor approximation.

Minor:

4. The phenomenon of loss spiking does not seem to be specific to low-order training, therefore, the proposed norm growth limiter is not intended for low-rank training.

5. The proposed method is only applicable to adaptive optimizers. I know that Adam is very important for efficient training, so this is not a major concern.

**Questions:**

Please address the concerns in weakness. I am impressed with the performance of the proposed method. However, I am skeptical that the proposed method works in the way the author suggests, or if it works in any other way. I would be willing to increase my score if the authors address these concerns.

---

> ### Author Response · Authors · 2024-11-23
> **Response to Reviewer Vz4X (1/3)**
>
> Thank you for acknowledging the performance of our method. We have added source code in **Supplementary Material** for reproducibility and will provide the link to our GitHub repository upon acceptance. Below are our responses to the specific concerns you raised.
>
> >W1: The authors support the use of low-rank scaling factors to approximate full-rank scaling factors by demonstrating that the order of the average scaling factors of the weight matrix is highly similar. However, similarity in order is not equivalent to similarity in value. Therefore, the validity of this approximation method is questionable.
>
> A1: Thank you for pointing out this. To **further verify the validity of our approximation**, we have conducted **more comprehensive experiments** on the similarity of scaling factors between low-rank and full-rank LLM training in Appendix A.6:
> + We added additional **value-based metrics** of similarity beyond order-based metrics (e.g., cosine similarity, mean squared error (MSE), and Pearson's correlation coefficient).
> + We **expanded the scope of our experiments** to include all matrices of LLaMA models ranging from 60M to 1B and assessed the similarity of scaling factors averaged over 10,000 steps.
> + We additionally performed a **column-level quantitative similarity analysis**, to evaluate the effectiveness of our column-level fine-grained strategy for scaling factors.
>
> The results are as follows:
> Table r1: Spearman, Kendall, and Pearson correlation coefficient (p-value) at both matrix and column levels.
> | Size   | Matrix Level    |  |  | Column Level    |    | |
> |--------|---------------------------|---------------------------|---------------------------|---------------------------|---------------------------|--------------------------|
> |        | Spearman      | Kendall |Pearson   | Spearman     | Kendall     | Pearson |
> | 60M    | 0.9972 (2e-62)            | 0.9662 (7e-26)            | 0.9891 (1e-46)| 0.9372 (0.0)              | 0.7942 (0.0)             |0.8723 (0.0) |
> | 130M   | 0.9925 (2e-76)            | 0.9409 (9e-37)            |0.9813 (2e-60) |0.8698 (0.0)              | 0.6830 (0.0)             | 0.7805 (0.0)|
> | 350M   | 0.9770 (3e-113)           | 0.8848 (5e-65)           | 0.9766 (1e-112)|0.9091 (0.0)              | 0.7400 (0.0)             |0.8272 (0.0) |
> | 1B     | 0.9469 (1e-83)            | 0.8249 (1e-56)           |0.9457 (6e-83) |0.8331 (0.0)              | 0.6513 (0.0)             |0.8112 (0.0) |
>
> Table r2:  Cosine Similarity and MSE at both matrix and column levels.
> | Size   | Matrix Level    |  | Column Level    | |
> |--------|---------------------------|---------------------------|---------------------------|---------------------------|
> |        | Cosine Similarity    | MSE     | Cosine Similarity   | MSE |
> | 60M    | 0.9922 | 3e-04 | 0.9273 | 3e-05 |
> | 130M   | 0.9901 | 2e-04 | 0.9046 | 2e-05 |
> | 350M   | 0.9893 | 1e-04 | 0.9174 | 1e-05 |
> | 1B     | 0.9795 | 2e-04 | 0.9229 | 1e-05 |
>
> Thus, we can observe significant similarity in scaling factor between low-rank and full-rank LLM training at **both order and value levels** across **a broad range of scenarios** (all coefficient and cosine similarity close to 1, while p-value and MSE close to 0). This insight provides **a robust experimental basis** for our proposed norm-based scaling in Fira and helps explain **the validity of this approximation**.

---

> ### Author Response · Authors · 2024-11-23
> **Response to Reviewer Vz4X (2/3)**
>
> >W3: If the scaling factor for the full rank gradient can be approximated by a low rank scaling factor, why not just apply the low rank scaling factor to the full rank gradient? This strategy has the same memory usage in the optimizer state as the proposed method. Could the authors conduct experiments to validate this strategy? This would be a good support for the effectiveness of the scaling factor approximation.
>
> A3: Thanks for your valuable suggestion. As you said, these experiments can be a good support for the effectiveness of the scaling factor approximation. Since it can also help us explain the second weakness, so we chose to address it in advance.
>
> We have conducted experiments to validate this strategy (**apply the low-rank scaling factors to the full-rank gradients**) by changing the Eq. (10) from $  W_{t+1}=W_{t} - \eta P_t\psi_t(R_t) - \eta \phi_t(R_t)  (G_t-P_tR_t)$ to $ W_{t+1}=W_{t} - \eta \phi_t(R_t)G_t$, i.e., **Fira-only-scaling**. In this way, we are able to **exclude the influence of the original Adam term** and better analyze the effectiveness of the scaling factor approximation. The results are as follows:
>
> Table r3:  Validation perplexity ($\downarrow$) across different ranks. **Fira-only-scaling** is the method mentioned above.
> | Rank |   4  |  16  |  64  |  128 |
> |------|------|------|------|------|
> | Fira-only-scaling | 35.91 | 32.90 | 31.93 | 31.68 |
> | Fira | 35.62 | 32.59 | 31.39 | 31.06  |
> | Galore | 73.35 | 56.57 | 40.22 | 34.88 |
> | Full-Rank (Adam) | 34.06 | | | |
>
> As shown in Table r3, the effectiveness of Fira consists of two parts, the Adam term and the scaling factor approximation. The higher the rank, the smaller the error of the approximation. Moreover, the gap between Fira-only-scaling and the original Fira is very small, indicating the effectiveness of the approximation.
>
> >W2: Since the proposed method aims to approximate full-rank training, we would assume that the performance of full-rank training is the upper bound. However, the proposed method even outperforms full-rank training. It is unclear why the proposed method outperforms.
>
> A2: Thanks for your valuable question. As the results in Table r3, if we only consider the effectiveness of rank value on approximation, full-rank training should be the upper bound of performance. As shown in Table r3, the effectiveness of **Fira-only-scaling** increases with rank growth, demonstrating the positive effect of increasing rank on approximation.
>
> However, compared to the full-rank Adam baseline which is **parameter-level** adaptive, Fira applies adaptive strategy at **matrix-level**, while **maintaining the original gradient direction** within a matrix (i.e., $\phi_t(R_t)G_t$) like SGD. As a result, the gradient direction which is only determined by the current state can introduce a **higher degree of randomness** in training. This randomness can enhance the model’s ability to **escape the local optima**, thus leading to **better generalization performance** [1,2,3]. This insight explains why Fira could match or even surpass the full-rank Adam baseline. On the other hand, despite inferior generalization performance, the adaptive method Adam is widely used for LLM training for its rapid convergence and high stability in LLM training [4]. Therefore, in addition to memory efficiency, Fira can also be seen as an innovative attempt to balance the advantages of Adam and SGD by the adaptive strategy at matrix level.
>
>
> [1]. Towards theoretically understanding why sgd generalizes better than adam in deep learning, In Neurips 20. \
> [2]. Understanding the Generalization of Adam in Learning Neural Networks with Proper Regularization, In ICLR 23. \
> [3]. Improving Generalization Performance by Switching from Adam to SGD, ArXiv.1712.07628 \
> [4]. Why are Adaptive Methods Good for Attention Models?, NeurIPS 2020

---

> ### Author Response · Authors · 2024-11-23
> **Response to Reviewer Vz4X (3/3)**
>
> >W4: The phenomenon of loss spiking does not seem to be specific to low-order training, therefore, the proposed norm growth limiter is not intended for low-rank training.
>
> A4: Thank you for pointing out this. As you said, there are a lot of methods also proposed to solve loss spiking and stabilize training, e.g., embedding normalization [5], gradient shrink [6], and tensor-wise scaling [7]. However, it is crucial to clarify that our Fira **does not conflict with these stabilization methods**. For instance, training the LLaMA model with Fira inherently incorporates stabilization methods like RMSNorm. Our norm-growth limiter is mainly aimed at **addressing the gradient stability capability that our norm-based scaling method lacks compared to Adam**. We have added additional ablation experiments in the Appendix. As shown in Figure 9, when we directly use Adam to pre-train the llama model, **there will be no loss spike**. However, since Fira maintains the original direction of the raw gradient $(G_t-P_tR_t)$, similar to SGD, it may lack the capability to navigate the sharp loss landscapes in LLM training, thus leading to **an additional loss spike**.
>
> Moreover, for further validation of the effectiveness of our limiter, we designed two additional variants to solve the loss spike: Gradient Shrink ($|S_t|=|S_t| \cdot \alpha+|S_{t-1}|\cdot(1-\alpha)$), and Tensor-Wise Scaling ($|S_t|=|S_t|\cdot \alpha $), where $S_t = \phi_t(R_t)  (G_t-P_tR_t) $  is the corrected gradient by applying our norm-based scaling.  The results are as follows:
>
> Table r4: Validation perplexity ($\downarrow$) across different gradient stabilization methods.
> | Norm-Growth Limiter |  Gradient Shrink  |  Tensor-Wise Scaling | Gradient Clipping  |  Without Limiter  |
> |------|------|------|------|------|
> |  **31.06** | 33.98  |  33.81 | 31.22  |  32.22 |
>
> As shown in Table r4, Fira outperforms other gradient stabilization methods. For further analysis, as shown in Figure 10 in Appendix, Gradient Shrink failed to solve the loss spike, while Tensor-Wise Scaling solved the loss spike but led to sub-optimal results.
>
> Overall, our limiter is designed to complement our norm-based scaling method, which can be compatible with other gradient stabilization methods.
>
> [5]. What language model to train if you have one million GPU hours?, EMNLP 2022
>
> [6]. GLM-130b: An open bilingual pre-trained model, In ICLR 2023
>
> [7]. 8-bit optimizers via block-wise quantization, In ICLR 2022
>
> >W5: The proposed method is only applicable to adaptive optimizers. I know that Adam is very important for efficient training, so this is not a major concern.
>
> A5: Thank you for pointing out this. As you said, adaptive optimizers such as Adam are commonly employed in LLM training. Given this, the substantial memory demands of their optimizer states present significant challenges. For example, the LLaMA 7B model,  cannot be directly full-rank trained on an A100 with 80G memory due to its large optimizer states. Consequently, for training models of 7B or larger, techniques such as DeepSeed Zero are often used to store the optimizer state in a distributed way, which increases significant additional communication overhead.
>
> However, Fira can solve the above problems by greatly reducing the optimizer's state memory usage while maintaining excellent training performance. It significantly reduces the memory bottleneck for LLM training while also avoiding additional communication overhead like DeepSeed technology.
>
> Lastly, we express our sincere gratitude for your insightful comments. Regarding the presentation issues raised, we will carefully incorporate all suggested improvements in our next revision. We hope that you will consider increasing your score if your concerns are adequately addressed. If you have any question for our paper, please feel free to point out and we will try to address it quickly.

---

> ### Comment · Reviewer_Vz4X · 2024-11-25
> **Thanks for your response.**
>
> Thank you for your detailed response. The empirical results of Fira-only-scaling are indeed impressive. The performance of Fira-only-scaling closely aligns well with that of Fira, supporting the approximation $\phi_t(R_t) \approx \phi_t(G_t)$. However, I have several follow-up questions and suggestions:
> 1. The Fira-only-scaling method, which is akin to full-rank gradient descent but with an adaptive scaling factor $\phi_t(R_t)$, outperforms Full-Rank (Adam) training for ranks greater than 16. This suggests that the scaling method is effective, yet it requires further investigation into how and why this scaling factor enhances performance.
>
> 2. As the performance of the Fira-only-scaling method improves with higher ranks, it suggests that the relationship $\phi_t(R_t) \approx \phi_t(G_t)$ becomes more accurate. This observation leads to the question of whether treating these as approximately equivalent might be too simplistic, especially at lower ranks.
>
> 3. Considering the performance improvements observed with higher ranks, it would be insightful to experiment with ranks closer to the full rank, such as 256-4, 256-16, and 256-64. These experiments could deepen our understanding of how Fira and Fira-only-scaling achieve their excellent results.
>
> 4. Lastly, the experimental settings mentioned in your response could be clarified. For example, the ranks used in Tables r1 and r2 are not specified, and Table r3 lacks information on the full rank. Providing these details would enhance the clarity and reproducibility of your results.
>
> In short, I suggest using Fira-scaling-only to verify the effectiveness of the proposed scaling factor $\phi_t(R_t)$ and show the experimental results with the rank variation at high ranks.

---

> ### Author Response · Authors · 2024-11-28
> **Thanks for your valuable feedback**
>
> Thanks for your valuable feedback. Below are our responses to the specific concerns you raised.
>
> >Q1: The Fira-only-scaling method, which is akin to full-rank gradient descent but with an adaptive scaling factor $\phi_t(R_t)$, outperforms Full-Rank (Adam) training for ranks greater than 16. This suggests that the scaling method is effective, yet it requires further investigation into how and why this scaling factor enhances performance.
>
> A1: Thank you for pointing out this. For further investigation, we have added additional experiments on the perplexity ($\downarrow$) trends among Fira, Fira-only-scaling, and Adam in Appendix A.11 on page 25 of the updated manuscript. As depicted in Figure 15 (b) and (c), while Adam demonstrates faster convergence during the initial stages, both Fira and Fira-only-scaling achieve superior performance in the later stages. This may be because Fira and Fira-only-scaling apply an adaptive strategy only at the matrix-level while maintaining the original gradient direction within a weight matrix. In this way, they may introduce a higher degree of randomness in training and a better ability to escape the local optima, thus achieving comparable or superior generalization performance.
>
> >Q2: As the performance of the Fira-only-scaling method improves with higher ranks, it suggests that the relationship $\phi_t(R_t) \approx \phi_t(G_t)$ becomes more accurate. This observation leads to the question of whether treating these as approximately equivalent might be too simplistic, especially at lower ranks.
>
>
> A2: Thank you for pointing out this. Indeed, due to the approximation error of $\phi_t(R_t) \approx \phi_t(G_t)$, the performance of Fira-only-scaling decreases as the rank value decreases. However, it is important to clarify that the memory savings from lower rank values are more critical than the corresponding performance degradation. As depicted in Table r3, Fira-only-scaling and Fira experience very little performance degradation and are far better than Galore. Furthermore, as illustrated in Figure 5 of our paper, Fira ($r/d_{model}=64/4096$) also demonstrates a significant improvement over GaLore ($r/d_{model}=512/4096$) for pre-training LLaMA 7B, while using 8× smaller rank and  memory usage for optimizer states. This demonstrates that the approximation of $\phi_t(R_t) \approx \phi_t(G_t)$ is still effective at lower ranks.
>
> >Q3: Considering the performance improvements observed with higher ranks, it would be insightful to experiment with ranks closer to the full rank, such as 256-4, 256-16, and 256-64. These experiments could deepen our understanding of how Fira and Fira-only-scaling achieve their excellent results.
>
> >Q4: Lastly, the experimental settings mentioned in your response could be clarified. For example, the ranks used in Tables r1 and r2 are not specified, and Table r3 lacks information on the full rank. Providing these details would enhance the clarity and reproducibility of your results.
>
> A3 & A4: Thanks for your valuable suggestion. We have added additional experiments with rank closer to the full rank (256-16) in Table r4. Additionally, the ranks presented in Tables r1 and r2 are consistent with those used in Table 2 of our manuscript. These ranking settings follow the pre-training benchmark in Galore. We have mentioned it in line 1117, Page 21 of the updated manuscript. Meanwhile, we have completed the full rank information in Table r4 (Fira and Galore can be viewed as Adam at full-rank condition, since $G_t - P_tR_t=0$).
>
> Table r4:  Validation perplexity ($\downarrow$) across different ranks.
> | Rank |   4  |  16  |  64  |  128 |   240   |
> |------|------|------|------|------|------|
> | Fira-only-scaling | 35.91 | 32.90 | 31.93 | 31.68 |  31.27|
> | Fira | 35.62 | 32.59 | 31.39 | 31.06  | 30.73 |
> | Galore | 73.35 | 56.57 | 40.22 | 34.88 | 32.00|
> |------|------|------|------|------|------|
> | Adam (Full-Rank)  | 34.06 | | | | |
> |Fira-only-scaling  (Full-Rank) | 30.84| | | | |
>
> As depicted in Table r4, Fira-scaling-only has also demonstrated its effectiveness with high ranks.

---

> ### Comment · Reviewer_Vz4X · 2024-11-28
> **Thanks for your response.**
>
> I thank the authors for their clarification and additional experimental results that confirm the effectiveness of the proposed scaling factor strategy. While it remains to explore further how the proposed scaling factor outperforms the one in Adam, it does work, even outside the scope of low-rank training! I have improved the score accordingly.

---

> ### Author Response · Authors · 2024-11-28
>
> Thank you for raising the score! We will further explore it in the future.

---

### Official Review · Reviewer_JS7v · 2024-11-04

**Soundness:** 3
**Presentation:** 3
**Contribution:** 3
**Rating:** 6
**Confidence:** 4

**Summary:**

This work proposes a new memory-efficient training approach built on top of GaLore. It leverages low-rank gradients while appropriately combining the residual gradient components with gradient scaling. Experiments demonstrates FIRA achieves significant improvements comparied with previous baselines in both pre-training and fine-tuning scenarios.

**Strengths:**

- The improvements is significant as shown in Table 2.

- The observations of gradient scaling, that can effectively incorporate the gradient residual is interesting and novel.

- The method is clearly explained and easy to follow.

**Weaknesses:**

- Intuitively, incorporating the residual gradient can lead to performance improvements, though it is typically not expected to surpass Adam. However, as shown in Table 2, Fira significantly outperforms Adam. Is there any insights why it is even better than adam.

- $\gamma = 1.01$ seems to effectively avoid the gradient spiking, whether the performance is senstive to the choice of $\gamma$.

**Questions:**

Please refer to the weakness

---

> ### Author Response · Authors · 2024-11-23
> **Response to Reviewer JS7v (1/2)**
>
> Thank you for acknowledging our method. We have added source code in **Supplementary Material** for reproducibility and will provide the link to our GitHub repository upon acceptance. Below are our responses to the specific concerns you raised.
>
> > W1: Intuitively, incorporating the residual gradient can lead to performance improvements, though it is typically not expected to surpass Adam. However, as shown in Table 2, Fira significantly outperforms Adam. Is there any insights why it is even better than adam.
>
> A1: Thank you for your valuable question. Compared to the full-rank Adam baseline which is **parameter-level** adaptive, Fira applies adaptive strategy at **matrix-level**, while **maintaining the original gradient direction** within a matrix (i.e., $\phi_t(R_t)G_t$) like SGD. As a result, the gradient direction which is only determined by the current state can introduce a **higher degree of randomness** in training. This randomness can enhance the model’s ability to **escape the local optima**, thus leading to **better generalization performance** [1,2,3]. This insight explains why Fira could match or even surpass the full-rank Adam baseline. On the other hand, despite inferior generalization performance, the adaptive method Adam is widely used for LLM training for its rapid convergence and high stability in LLM training [4]. Therefore, in addition to memory efficiency, Fira can also be seen as an innovative attempt to balance the advantages of Adam and SGD by the adaptive strategy at matrix level.
>
> [1]. Towards theoretically understanding why sgd generalizes better than adam in deep learning, In Neurips 20. \
> [2]. Understanding the Generalization of Adam in Learning Neural Networks with Proper Regularization, In ICLR 23. \
> [3]. Improving Generalization Performance by Switching from Adam to SGD, ArXiv.1712.07628 \
> [4]. Why are Adaptive Methods Good for Attention Models?, NeurIPS 2020

---

> > ### Comment · Reviewer_JS7v · 2024-11-25
> > **Thanks for the responses**
> >
> > Thanks for your responses. The ablation study of $\gamma$ appears satisfactory to me. While the current state gradient in Fira introduces randomness in training, aiding in escaping local optima, this benefit could also be achieved with SGD? However, since SGD is generally considered unsuitable for training transformers [1], could you elaborate more on why FiRA is preferable to full-rank Adam?
> >
> > [1] Why Transformers Need Adam: A Hessian Perspective

---

> ### Author Response · Authors · 2024-11-23
> **Response to Reviewer JS7v (2/2)**
>
> >  $\gamma = 1.01$ seems to effectively avoid the gradient spiking, whether the performance is senstive to the choice of $ \gamma $.
>
> A2: Thanks for your valuable suggestion. We have conducted ablation experiments of $\gamma$. The results are as follows:
>
> Table r1: Validation perplexity ($\downarrow$) of Fira across different choices of $ \gamma $.
> | $\gamma$ | $\infty$ (w.o. limiter) | 1.1 | 1.01 | 1.001 | 1 |
> |------|------|------|------|------|------|
> | PPL($\downarrow$) | 32.22 | 32.09 | **31.06** | 31.26 | 31.28 |
>
> As shown in Table r1, the performance of Fira is **not sensitive** to the choice of $ \gamma $. Provided that $ \gamma \le 1.01 $, Fira can **effectively mitigate spikes in loss**, with only a marginal decrease in performance when $\gamma=1$. As we can see, the setting of $\gamma = 1.01$, employed across all experiments in this paper, is highly effective. This value is neither too small to limit the normal growth of the gradients, nor too large to limit the sudden increase of the gradients.
>
> In addition, for more comprehensive comparisons of our norm-growth limiter, we designed two additional gradient stabilization variants to solve the loss spike: Gradient Shrink ($|S_t|=|S_t| \cdot \alpha+|S_{t-1}|\cdot(1-\alpha)$), and Tensor-Wise Scaling ($|S_t|=|S_t|\cdot \alpha $), where $S_t = \phi_t(R_t)  (G_t-P_tR_t) $  is the corrected gradient by applying our norm-based scaling. The results are as follows:
>
> Table r2: Validation perplexity ($\downarrow$) across different gradient stabilization methods.
> | Norm-Growth Limiter |  Gradient Shrink  |  Tensor-Wise Scaling | Gradient Clipping  |  Without Limiter  |
> |------|------|------|------|------|
> |  **31.06** | 33.98  |  33.81 | 31.22  |  32.22 |
>
> As shown in Table r2, Fira outperforms other gradient stabilization methods. For further analysis, as shown in the additional Figure 10 in Appendix, Gradient Shrink failed to solve the loss spike, while Tensor-Wise Scaling solved the loss spike but led to sub-optimal results.
>
> Lastly, we express our sincere gratitude for your insightful comments. Regarding the presentation issues raised, we will carefully incorporate all suggested improvements in our next revision. We hope that you will consider increasing your score if your concerns are adequately addressed. If you have any question for our paper, please feel free to point out and we will try to address it quickly.

---

> ### Author Response · Authors · 2024-11-28
> **Thanks for your valuable feedback**
>
> Thanks for your valuable feedback. As you mentioned, SGD is generally considered unsuitable for training transformers because it **applies the same learning rate to all weight blocks**, which overlooks their inherent differences [1]. As a result, the benefit of escaping local optima can't be directly achieved with SGD in LLM training. However, while maintaining the original gradient direction within each block, Fira applies adaptive learning rates to different weight blocks based on their scaling factors. In this way, Fira can expand the benefit of achieving comparable or superior generalization performance in SGD to LLM training.
>
> For further validation, we have added additional experiments on the perplexity ($\downarrow$) trends among Fira, SGD, and Adam in Appendix A.11 on page 25 of the updated manuscript. As illustrated in Figure 25 (a), the performance of vanilla SGD is significantly inferior, highlighting its inadequacy for directly training LLMs. As depicted in Figure 15 (b), while Adam demonstrates faster convergence during the initial stages, Fira achieves superior performance in the later stages due to the randomness introduced by the current state gradient.
>
> [1]. Why Transformers Need Adam: A Hessian Perspective.

---

### Official Review · Reviewer_R8Z3 · 2024-11-04

**Soundness:** 2
**Presentation:** 3
**Contribution:** 2
**Rating:** 5
**Confidence:** 4

**Summary:**

The paper proposes a memory-efficient training framework called Fira for the pre-training and fine-tuning of large language models (LLMs). Fira aims to reconcile the memory efficiency of low-rank training with the performance benefits of full-rank updates. Building upon GaLore [1], Fira introduces two key components: Norm-Based Scaling and the Norm-Growth Limiter. In Norm-Based Scaling, Fira leverages the empirical observation that the scaling effect on gradient norms remains similar between low-rank and full-rank training. Thus, it uses scaling factors from low-rank optimizers to approximate gradient corrections in full-rank training. The Norm-Growth Limiter mitigates sudden spikes in training loss by limiting the relative increase in gradient norms between steps, effectively converting sudden spikes into gradual rises. The paper validates Fira in both pre-training by evaluating the perplexity on the C4 dataset and fine-tuning by evaluating performance on the CommonSenseQA tasks across various model sizes (60M, 130M, 350M, 1B).

[1] GaLore: Memory-Efficient LLM Training by Gradient Low-Rank Projection

**Strengths:**

1. The observation of the similarity in scaling factors of adaptive optimizers between low-rank and full-rank training is insightful and forms a solid foundation for the proposed method.
2. In Norm-Based Scaling, Fira not only maintains memory efficiency but also enables full-rank updates, giving it an advantage over other methods.
3. The Norm-Growth Limiter is a simple yet effective approach to addressing sudden spikes in training loss.
4. The authors conduct extensive experiments, including ablation studies and scaling analyses, across multiple model sizes and tasks, strengthening the empirical validation of their approach.

**Weaknesses:**

1. Limited theoretical justification. While the empirical observations are compelling, the paper lacks a rigorous theoretical analysis explaining why norm-based scaling effectively approximates full-rank gradient corrections.
2. Inconsistent and incomplete baseline comparisons. The baseline comparisons are not comprehensive enough; more recent memory-efficient fine-tuning methods (such as LISA[2], FLoRA[3] that gradient based memory-efficient training method) should be included for a stronger evaluation. Additionally, the evaluation settings between pre-training and fine-tuning are inconsistent: in pre-training, the comparisons include full-rank, Fira, GaLore, LoRA, and ReLoRA, whereas in fine-tuning, full-rank and ReLoRA are omitted, and instead, prefix tuning, series, and parallel methods are included. This inconsistency leads to some confusion. Since the scope of comparison is memory-efficient training, the experiments should be as consistent as possible.
3. Lack of comparison with existing methods addressing loss spikes. Restricting the magnitude of gradients to address abrupt changes in gradients or loss is a commonly adopted approach. While gradient clipping employs absolute value thresholds, the Norm-Growth Limiter uses a relative threshold. Furthermore, sudden spikes are not unique to memory-efficient or parameter-efficient training; they are common issues even in full training. Several studies have proposed methods based on gradient and layer norms to address loss spikes [4,5,6]; however, the paper does not provide comparisons with these methods.
4. Lack of detailed memory efficiency metrics. As a memory-efficient training method, the paper lacks actual memory usage comparisons. In the comparison with GaLore, the paper emphasizes training performance but overlooks memory efficiency and throughput, which is inconsistent with the experiments provided by GaLore.

[2] LISA: Layerwise Importance Sampling for Memory-Efficient Large Language Model Fine-Tuning
[3] Flora: Low-Rank Adapters Are Secretly Gradient Compressors
[4] What language model to train if you have one million GPU hours
[5] GLM-130b: An open bilingual pre-trained model
[6] 8-bit optimizers via block-wise quantization

**Questions:**

1. Can you provide theoretical analysis explaining why the norm-based scaling effectively approximates full-rank gradient corrections? Is this phenomenon present in any LLM training, or is it conditional?
2. Can the baseline comparisons in pre-training and fine-tuning be kept consistent?
3. As a memory-efficient training method, could you provide comparisons regarding training memory usage? Including more experiments on overhead would be a plus.
4. For the Norm-Growth Limiter, could you provide comparative experiments with other methods that address loss spikes?

---

> ### Author Response · Authors · 2024-11-23
> **Response to Reviewer R8Z3 (1/4)**
>
> Thanks for the reviewer's detailed feedback and comments. Below are our responses to the specific concerns you raised.
>
> >W1 & Q1: Can you provide theoretical analysis explaining why the norm-based scaling effectively approximates full-rank gradient corrections? Is this phenomenon present in any LLM training, or is it conditional?
>
> A1: Thanks for your valuable suggestion. To quantify the effectiveness of our approximates in norm-based scaling, we have added **theoretical analysis of error upper bound** on this approximation in Appendix A.1, and **verified our analysis experimentally**. Moreover, we have conducted **more comprehensive experiments** on the similarity of scaling factors between low-rank and full-rank training in Appendix A.6 to confirm that this phenomenon is **widespread in LLM training**.
>
> **Theoretical error upper bound on the approximation of norm-based scaling.**
>
> In the paper, we use the scaling factors of low-rank gradients to approximate that of full-rank gradients. To quantify the effectiveness of this approximation, we derived its error upper bound theoretically:
>
> $$ \left| \phi^2_t(G_t)-\phi^2_t(R_t)  \right| \leq (c_{\max}^2 - c_{\min}^2) \cdot (1-\frac{\sum_{i=1}^{r} g_i^2}{\sum_{i=1}^{n} g_i^2}),$$
>
> where $\phi_t(G_t)$, $\phi_t(R_t)$ are the scaling factors of full-rank and low-rank gradients respectively. We assume that scaling factors $\phi_t$ are bounded between constants $c_{\max}$, $c_{\min}$, and $g_i$ is the component of the gradients $g$. To simplify the proof, we consider that $r$ components of low-rank gradients are directly sampled from $n$ components of full-rank gradients. From this theory, we can find that the error upper bound on the approximation of scaling factors is mainly determined by two aspects:
>
> + Variability of Scaling Factor $(c_{\max}^2 - c_{\min}^2)$: This term represents the maximum variation in the scaling factors of different gradient components. For further validation, we designed **Fira-only-scaling**, a variant of Fira. It directly applies the low-rank scaling factors to the full-rank gradients by changing the Eq. (10) from $  W_{t+1}=W_{t} - \eta P_t\psi_t(R_t) - \eta \phi_t(R_t)  (G_t-P_tR_t)$ to $ W_{t+1}=W_{t} - \eta \phi_t(R_t)G_t$. In this way, we are able to exclude the influence of the original Adam term $P_t\psi_t(R_t)$ and better analyze the effectiveness of our approximation. As shown in Table r1, Fira (column-level) gains better performance than Fira (matrix-level) for its more fine-grained consideration of the scaling factor, which also means a smaller maximum variation $(c_{\max}^2 - c_{\min}^2)$.
>
> + Effectiveness of Gradient Sampling $(1-\frac{\sum_{i=1}^{r} g_i^2}{\sum_{i=1}^{n} g_i^2})$: This term represents the proportion of the gradients norm contributed by the sampled low-rank \( r \) components from full-rank \( n \) components. As shown in Table r2, we conducted ablation experiments **Fira-only-scaling-w.o.-svd**, i.e., Fira-only-scaling without SVD in low-rank gradient sampling.  As we can see, SVD is capable of sampling more prominent low-rank gradients, which leads to a reduction in the upper bound of error and enhanced performance. Similarly, as shown in Table r3, employing a higher rank enables the sampling of a greater proportion of the gradients norm, resulting in reduced error upper bound and improved performance.
>
> Table r1: Validation perplexity ($\downarrow$)  of Fira-only-scaling at Column-level and Matrix-level.
> | Column-level | Matrix-level |
> |------|------|
> | **31.68** | 32.05 |
>
> Table r2: Validation perplexity ($\downarrow$)  of Fira-only-scaling with and without SVD.
> | Fira-only-scaling | Fira-only-scaling-w.o.-svd |
> |------|------|
> |**31.68** | 32.22 |
>
> Table r3:  Validation perplexity ($\downarrow$) of Fira-only-scaling across different ranks.
> | Rank |   4  |  16  |  64  |  128 |
> |------|------|------|------|------|
> | Fira-only-scaling | 35.91 | 32.90 | 31.93 | **31.68** |

---

> ### Author Response · Authors · 2024-11-23
> **Response to Reviewer R8Z3 (2/4)**
>
> **Continuation of A1.**
>
> **Widespread presence of the phenomenon of scaling factor similarity.**
>
> We conducted **more comprehensive experiments** on the similarity of scaling factors between low-rank and full-rank LLM training.
> + We added additional **value-based metrics** of similarity beyond order-based metrics (e.g., cosine similarity, mean squared error (MSE), and Pearson's correlation coefficient).
> + We **expanded the scope of our experiments** to include all matrices of LLaMA models ranging from 60M to 1B and assessed the similarity of scaling factors averaged over 10,000 steps.
> + We additionally performed a **column-level quantitative similarity analysis**, to evaluate the effectiveness of our column-level fine-grained strategy for scaling factors.
>
> The results are as follows:
>
> Table r4: Spearman, Kendall, and Pearson correlation coefficient (p-value) at both matrix and column levels.
> | Size   | Matrix Level    |  |  | Column Level    |    | |
> |--------|---------------------------|---------------------------|---------------------------|---------------------------|---------------------------|--------------------------|
> |        | Spearman      | Kendall |Pearson   | Spearman     | Kendall     | Pearson |
> | 60M    | 0.9972 (2e-62)            | 0.9662 (7e-26)            | 0.9891 (1e-46)| 0.9372 (0.0)              | 0.7942 (0.0)             |0.8723 (0.0) |
> | 130M   | 0.9925 (2e-76)            | 0.9409 (9e-37)            |0.9813 (2e-60) |0.8698 (0.0)              | 0.6830 (0.0)             | 0.7805 (0.0)|
> | 350M   | 0.9770 (3e-113)           | 0.8848 (5e-65)           | 0.9766 (1e-112)|0.9091 (0.0)              | 0.7400 (0.0)             |0.8272 (0.0) |
> | 1B     | 0.9469 (1e-83)            | 0.8249 (1e-56)           |0.9457 (6e-83) |0.8331 (0.0)              | 0.6513 (0.0)             |0.8112 (0.0) |
>
> Table r5:  Cosine Similarity and MSE at both matrix and column levels.
> | Size   | Matrix Level    |  | Column Level    | |
> |--------|---------------------------|---------------------------|---------------------------|---------------------------|
> |        | Cosine Similarity    | MSE     | Cosine Similarity   | MSE |
> | 60M    | 0.9922 | 3e-04 | 0.9273 | 3e-05 |
> | 130M   | 0.9901 | 2e-04 | 0.9046 | 2e-05 |
> | 350M   | 0.9893 | 1e-04 | 0.9174 | 1e-05 |
> | 1B     | 0.9795 | 2e-04 | 0.9229 | 1e-05 |
>
> From the above results, we can observe the **significant similarity** of scaling factors between low-rank and full-rank LLM training (all coefficient and cosine similarity close to 1, while P-value and MSE close to 0), manifesting across a broad range of scenarios. This insight provides a robust experimental basis for our proposed norm-based scaling in Fira and confirms that this phenomenon is **widespread in LLM training**.

---

> ### Author Response · Authors · 2024-11-23
> **Response to Reviewer R8Z3 (3/4)**
>
> > W2 & Q2: Can the baseline comparisons in pre-training and fine-tuning be kept consistent?
>
> A2: Thanks for your valuable suggestion. Since the benchmarks for pre-training and fine-tuning differ, the evaluation settings and baselines are accordingly distinct. We follow previous works to evalute the pre-training task [1] and the fine-tuning task [2].
> To ensure greater consistency, we have incorporated additional baselines as follows.
>
> + **More comprehensive comparisons**. Due to rebuttal ddl, we have incorporated an additional memory-efficient training baseline, Flora [3]. Meanwhile, LISA [4] and Flora have been added to Related Work in our paper. We will add LISA baseline to our final version.
>
> + **More consistent comparisons**. In original Galore setting, the pre-trained RoBERTa-Base model was fine-tuned using the GLUE benchmark. However, since the number of parameters in this model is too small (**less than 1B**), we switched it to the LLM-Adapters [2] benchmark, which involves fine-tuning the LLaMA 7B model on 8 commonsense reasoning datasets. To address this, we included remaining full-rank and ReLoRA baselines in our fine-tuning experiments.
>
> The results are as follows:
>
> Table r6: Validation perplexity ($\downarrow$) for pre-training LLaMA 60M on C4 dataset.
> | Fira | Flora | Galore | LoRA | ReLoRA|Full-rank |
> |------|------|------|------|------|------|
> |  **31.06** | 38.77  |  34.88 |  34.99 |37.04 | 34.06|
>
> Table r7: Accuracy ($\uparrow$) for fine-tuning LLaMA 7B on eight commonsense reasoning datasets.
>
> | Method | BoolQ | PIQA | SIQA | HellaSwag |WinoGrande | ARC-e | ARC-c | OBQA  | Avg |
> |------|------|------|------|------|------|------|------|------|------|
> | Fira  |  69.4 | **82.6**|**78.0** |76.8 |**81.2** | **82.2**| 64.4| **80.8**| **76.9** |
> | Flora |  50.1 |77.5 |74.2 |53.8 |45.5 | 79| **64.6**| 74.8|64.9 |
> | ReLoRA|  68.9 | 81.2| 77.8| 46|79.4 | 80.2| 64.2| 79.6|72.2 |
> | LoRA  | 68.9  | 80.7| 77.4| **78.1**| 78.8| 77.8| 61.3| 74.8| 74.7|
> | Galore|  **69.5** |82.0 | 75.1| 32.2| 18.0| 80.7| 65.8| 78.0| 62.7|
> | Full-rank |  64.2 |68.1 | 68.0|42.3 | 66.5| 55.6| 43.9| 60.0| 58.6|
>
> As shown in Table r6 and r7, Fira outperforms our baseline methods for both the pre-training and fine-tuning benchmarks.
>
> [1]. GaLore: Memory-Efficient LLM Training by Gradient Low-Rank Projection, In ICML 2024
>
> [2]. LLM-Adapters: An Adapter Family for Parameter-Efficient Fine-Tuning of Large Language Models, In EMNLP 2023
>
> [3]. Flora: Low-Rank Adapters Are Secretly Gradient Compressors, In ICML 2024
>
> [4]. LISA: Layerwise Importance Sampling for Memory-Efficient Large Language Model Fine-Tuning, In NeurIPS 2024

---

> ### Author Response · Authors · 2024-11-23
> **Response to Reviewer R8Z3 (4/4)**
>
> > W4 & Q3: As a memory-efficient training method, could you provide comparisons regarding training memory usage? Including more experiments on overhead would be a plus.
>
> A3: Thanks for your valuable suggestion. We have added additional comparisons regarding **real memory usage** and **throughput** of different memory-efficient training methods for both pre-training and fine-tuning in Appendix A.9. The results are as follows:
>
> Table r8: Real memory usage and normalized throughput when pre-training LLaMA 1B on the C4 dataset.
> |      | Fira | Galore | Flora | LoRA | ReLoRA | Full-rank |
> |------|------|------|------|------|------|------|
> |   Memory (GB)   | 54.6 | 54.6 | 54.5 | 59.0 | 59.0 | 58.5 |
> |  Normalized Throughput (%)   | 94.2 |  95.9 |  95.9     |   67.4  | 67.4 |   100 |
>
> Table r9: Real memory usage and normalized throughput when fine-tuning LLaMA 7B on commonsense reasoning datasets.
> |      | Fira | Galore | Flora | LoRA | ReLoRA | Full-rank |
> |------|------|------|------|------|------|------|
> |   Memory (GB)   | 23.4 | 23.4 |23.3 |23.7 |23.7 | > 80 |
> |  Normalized Throughput (%)   |156.1 |  201.1 |  210.3  |   232.8  | 232.8 |   100 |
>
>
>
> As illustrated in Tables r8 and r9, Fira achieves superior memory efficiency compared to full-rank training without significantly reduced throughput. Although Fira's throughput is slightly lower than that of other memory-efficient methods, it delivers exceptional performance, as shown in the previous section. During pre-training, methods like LoRA necessitate maintaining higher-rank adapters compared to full-rank training. In practice, maintaining these higher-rank adapters outweighs the benefits of fewer trainable parameters, thus leading to more memory and less throughput. Furthermore, since full fine-tuning of LLaMA 7B's memory requirements exceeds the A100's 80GB capacity, we utilize DeepSeed's Zero2 technology to mitigate its memory usage.
>
> > W3 & Q4: For the Norm-Growth Limiter, could you provide comparative experiments with other methods that address loss spikes?
>
> A4: Thanks for your valuable suggestion. We have cited the paper provided [5,6,7] and added additional comparative experiments in Appendix A.7. We designed two additional variants to solve the loss spike: Gradient Shrink ($|S_t|=|S_t| \cdot \alpha+|S_{t-1}|\cdot(1-\alpha)$), and Tensor-Wise Scaling ($|S_t|=|S_t|\cdot \alpha $), where $S_t = \phi_t(R_t)  (G_t-P_tR_t) $  is the corrected gradient by applying our norm-based scaling. The results are as follows:
>
> Table r10: Validation perplexity ($\downarrow$) across different gradient stabilization methods.
> | Norm-Growth Limiter |  Gradient Shrink  |  Tensor-Wise Scaling | Gradient Clipping  |  Without Limiter  |
> |------|------|------|------|------|
> |  **31.06** | 33.98  |  33.81 | 31.22  |  32.22 |
>
> As shown in Table r10, Fira outperforms other gradient stabilization methods. For further analysis, as shown in the additional Figure 10 in Appendix, Gradient Shrink failed to solve the loss spike, while Tensor-Wise Scaling solved the loss spike but led to sub-optimal results. In addition, it is crucial to clarify that **Fira does not conflict with other gradient stabilization methods** such as **embedding normalization**. For instance, training the LLaMA model with Fira inherently incorporates stabilization methods like RMSNorm.
>
> Our norm-growth limiter is mainly aimed at addressing the gradient stability capability that our norm-based scaling method lacks compared to Adam. We have added additional ablation experiments in the Appendix. As shown in Figure 9, when we directly use Adam to pre-train the llama model, there will be no loss spike. However, since Fira maintains the original direction of the raw gradient $(G_t-P_tR_t)$, similar to SGD, it may lack the capability to navigate the sharp loss landscapes in LLM training, thus leading to an additional loss spike.
>
> Overall, our limiter is designed to complement our norm-based scaling method and is compatible with other gradient stabilization techniques.
>
>
> [5]. What language model to train if you have one million GPU hours?, EMNLP 2022
>
> [6]. GLM-130b: An open bilingual pre-trained model, In ICLR 2023
>
> [7]. 8-bit optimizers via block-wise quantization, In ICLR 2022
>
> Lastly, we express our sincere gratitude for your insightful comments. Regarding the presentation issues raised, we will carefully incorporate all suggested improvements in our next revision. We have added source code in **Supplementary Material** for reproducibility and will provide the link to our GitHub repository upon acceptance. We hope that you will consider increasing your score if your concerns are adequately addressed. If you have any question for our paper, please feel free to point out and we will try to address it quickly.

---

> > ### Comment · Reviewer_R8Z3 · 2024-11-28
> >
> > I appreciate the authors' effort on addressing my questions in the rebuttal. With the newly added theoretical and experimental results, I would raise my score to 5.

---

> ### Author Response · Authors · 2024-11-28
>
> Thank you for raising the score! It's glad to see your questions have been addressed!

---

### Meta-Review · Area_Chair_6vjd · 2024-12-26

**Metareview:**

This paper proposes a new strategy for training LLMs, where they use the full rank update but maintain optimizer stats in a low-rank fashion to save memory. Two main techniques: Norm-Based Scaling and the Norm-Growth Limiter are proposed to form the actual updates using low-rank stats. Initially reviewers have concerns about both the empirical comparison to previous methods and the theoretical justification of the proposed method. After the rebuttal and author-reviewer discussions, the former is mostly addressed and most of the reviewers agree that the paper demonstrates good results. However, the later (theoretical justification) was not fully addressed: even after rebuttal and discussions, reviewers still think that the paper requires a major revision to provide a more convincing analysis for the proposed method. Further, some reviewers pointed out that Norm-Growth Limiter is similar to gradient clipping which seems to be also used in other places and not specific to low-rank training. Due to these concerns, we decide to recommend rejection and encourage the authors to improve the justification of their proposed method based on the reviewers’ feedback.

**Additional Comments On Reviewer Discussion:**

During reviewer discussions, both reviewer qp5x and R8z3 are still having concerns about the theoretical justification of the paper. In particular reviewer qp5x suggested the following points for the authors to address in their revision:

“””
I recommend the authors to
Substantially rewrite the introduction part regarding how Fira is designed. Replace the rank-order similarity experiments Figure 3 with the magnitude deviation visualization in Figure 11, 12, 13, and 14.
Add theoretical or empirical analysis on how  holds under varying .
Add reasonable analysis on how Fira obtain extra performance gain compared to the full-rank baseline.
”””

---

### Decision · Program_Chairs · 2025-01-22

Reject